# NiCo identifies extrinsic drivers of cell state modulation by niche covariation analysis

Ankit Agrawal [1], Stefan Thomann [1], Sukanya Basu [1] & Dominic Grün [1,2] ✉

Cell states are modulated by intrinsic driving forces such as gene expression noise and extrinsic signals from the tissue microenvironment. The distinction between intrinsic and extrinsic cell state determinants is essential for understanding the regulation of cell fate in tissues during development, homeostasis and disease. The rapidly growing availability of single-cell resolution spatial transcriptomics makes it possible to meet this challenge. However, available computational methods to infer topological tissue domains, spatially variable genes, or ligand-receptor interactions are limited in their capacity to capture cell state changes driven by crosstalk between individual cell types within the same niche. We present NiCo, a computational framework for integrating single-cell resolution spatial transcriptomics with matched single-cell RNA-sequencing reference data to infer the influence of the spatial niche on the cell state. By applying NiCo to mouse embryogenesis, adult small intestine and liver data, we demonstrate the ability to predict novel niche interactions that govern cell state variation underlying tissue development and homeostasis. In particular, NiCo predicts a feedback mechanism between Kupffer cells and neighboring stellate cells dampening stellate cell activation in the normal liver. NiCo provides a powerful tool to elucidate tissue architecture and to identify drivers of cellular states in local niches.

Cellular states are marked by distinct molecular configurations which determine fate and function[1,2], and are intricately regulated by a complex interplay of myriad molecules, interacting stochastically, only biased by their affinities. Despite this complexity, cell states exhibit remarkable robustness ensuring reproducible cell fate decision and differentiation. Within tissues, cell state variability can be dissected into intrinsic and extrinsic determinants.

For instance, stochastic binding of transcription factors to regulatory sites may lead to strong fluctuations of mRNA molecules across cells within the same state. This transcriptional bursting represents a cell-intrinsic source of variability and could even be actively modulated and exploited for the regulation of developmental processes[3]. Extrinsic molecular determinants, on the other hand, arise from cell-cell communication in tissues, encompassing molecular signaling, physical interactions, and competition for metabolites. Cell-cell communication is pivotal for physiological processes such as development, homeostasis, disease progression, and regeneration. Deciphering the gene programs orchestrating spatiotemporal communication between cells within tissues remains an open challenge for single-cell biology[4].

The ability to dissect intrinsic and extrinsic determinants of a cell state would allow us to acquire a deeper understanding of cell state regulation within tissues. Although single-cell RNA-sequencing (scRNA-seq) permits the characterization of cell state heterogeneity and gene expression noise[5–7], and the prediction of ligand-receptor interactions[8,9], spatial context is lacking, and a distinction between intrinsic and extrinsic drivers of cell state changes is not possible.

Over the last decade, spatially resolved high-resolution transcriptomics technologies have rapidly advanced[10]. Following the development of sequencing-based spatial transcriptomics with a

[1]Würzburg Institute of Systems Immunology, Julius-Maximilians-Universität Würzburg, Würzburg, Germany. [2]CAIDAS - Center for Artificial Intelligence and Data Science, Würzburg, Germany. ✉e-mail: dominic.gruen@uni-wuerzburg.de

resolution of ~100 μm[11], several improved methods have been introduced increasing the resolution to the sub-micrometer range (e.g., Stereo-seq[12], Seq-Scope[13]). However, owing to the low sensitivity of transcript detection, pixels typically need to be aggregated into larger bins, thereby sacrificing single-cell resolution[10]. As a complementary approach, highly-multiplexed single-molecule FISH (smFISH), pioneered by MERFISH[14] and seqFISH[15], or in situ sequencing[16–18] rely on imaging-based quantification of individual mRNA molecules in tissue sections, typically for a few hundred genes. Crucially, the assignment of single mRNA molecules to specific cells is a prerequisite for measuring covariation of gene expression in neighboring cells to understand extrinsic divers of cell state changes, and can currently only be achieved by imaging-based methods allowing cell segmentation. Availability of commercial solutions for multiplexed smFISH and in situ sequencing is rapidly growing enabling the broader community to explore fundamental aspects of cell state regulation in tissues.

Extrinsic determinants of the transcriptional state of a cell originating from the microenvironment can be revealed by gene expression covariance in neighboring cells. We developed NiCo with the goal to infer *Ni*che *Co*variation of gene expression programs at cell type resolution on a transcriptome-wide scale by integrating imaging-based spatial transcriptomics with matched scRNA-seq reference data. Available computational approaches for the analysis of spatial transcriptomics are focused on the inference of spatially variable genes (e.g., SpatialDE[19]), on spatial modeling of ligand-receptor interactions (e.g., SpaOTsc[20] and COMMOT[21]) or local gene-gene dependencies (e.g., MISTy[22] or GCNG[23]), or on explaining intra-cell-type variance by niche composition (NCEM)[24]. These methods are typically restricted to the set of genes directly measured in the spatial modality, and none of these methods was designed towards inferring cell state dependencies of co-localized cell types. To overcome these limitations NiCo integrates sensitive genome-wide quantification of gene expression by scRNA-seq data with single-cell resolution spatial transcriptomics. NiCo does not rely on mapping individual cells from the scRNA-seq data into space or vice versa, which is computationally expensive and inherently noisy. Instead, NiCo derives interpretable latent factors reflecting cell state variability of each cell type captured within both scRNA-seq reference and spatial data. These latent factors are then used to infer covariation of gene programs in neighboring cell types from the spatial modality and to associate these factors with transcriptome-wide expression patterns by utilizing the scRNA-seq data. We demonstrate that this approach has the capacity to recover well-known functional niche interactions in the developing mouse embryo, as well as in mouse intestine and liver. NiCo provides novel insights into the cell states involved in these interactions and their spatial configuration. NiCo complements existing methods specialized for spatial ligand-receptor interaction analysis to infer cell state covariation in local niches and illuminates the consequences of local niche interactions for cell state control.

## Results

### Inferring covariation of gene programs in local tissue niches at single-cell resolution with NiCo

NiCo integrates imaging-based spatial transcriptomics data, obtained by multiplexed smFISH or in situ sequencing technologies such as MERFISH[14], seqFISH[15], or STARmap[18], and commercial implementations of these approaches (e.g., Vizgen MERSCOPE, 10x Xenium, Nanostring CosMx) which typically measure targeted libraries of up to 500 genes, with scRNA-seq reference data that provide genome-wide coverage but lack spatial information. NiCo requires a cell-by-gene count matrix and two-dimensional cell-center coordinates from imaging-based spatial transcriptomics data that have undergone cell segmentation, as well as a cell-by-gene count matrix of a scRNA-seq reference dataset along with cell type labels comprising all cell types expected to occur in the spatial data.

NiCo was designed as a multi-step pipeline, facilitating (1) cell type annotation of the spatial modality by label transfer from the scRNA-seq reference data, (2) recovery of niche architecture for each cell type, (3) inference of covarying latent factors in co-localized cell types, and association of these latent factors with functional pathways and molecular signaling interactions (Fig. 1).

NiCo employs an iterative approach to perform the cell type annotations (Fig. 1, "Annotations"). First, mutual nearest neighbors are identified in the spatial and scRNA-seq data modalities after normalization to eliminate technical variability. After pruning scattered anchors utilizing Leiden clusters of the spatial modality, non-anchors are iteratively annotated based on anchors among their k-nearest neighbors (Methods).

After annotating cell types in the spatial domain, NiCo interrogates niche architecture for each "central" cell type (CC) in order to identify neighboring niche cell types with a high predictive capacity for the central cell type identity. NiCo trains a regularized logistic regression classifier to predict the identity of the central cell type from the normalized cell type frequencies in local niche neighborhoods (Methods). The regression coefficients of all niche cell types permit prioritization of potential interaction partners within local neighborhoods (Fig. 1, "Interactions").

As a first step towards inferring covariation of gene programs within locally interacting cell types, NiCo has implemented different versions of non-negative matrix factorization (NMF). For each cell type, NiCo infers latent factors by integrative NMF[25,26] to capture common variability in the spatial and the scRNA-seq data modality on the shared subset of genes. Alternatively, if gene expression variability in the spatial modality is dominated by technical artefacts, e.g., due to imperfect segmentation giving rise the "spill-over" between cell types, NiCo offers conventional NMF on the shared set of genes only for the scRNA-seq modality and infers the cell loadings of these factors in the spatial modality. The output of this step is a set of latent factors capturing the cell state variability of each cell type based on the shared set of genes between both modalities (Fig. 1, "Covariations").

Finally, NiCo performs ridge regression of each latent factor of the central cell on the latent factors of all predictive niche cell types obtained by step 2 ("Interactions"). Significant regression coefficients indicate either positive or negative covariation between the respective factors (Fig. 1, "Covariations"). Functional gene modules can be associated with covarying factors by performing pathway enrichment analysis on the top (anti-)correlating genes for each factor obtained from the transcriptome-wide scRNA-seq data. In addition, these genes can be interrogated for covarying ligands and receptors as potential mediators of this crosstalk.

We note that NiCo can also be applied to genome-wide sequencing- or imaging-based spatial transcriptomics data without requiring scRNA-seq reference data. In this case, the "Interactions" and "Covariations" modules can directly be run on annotated spatial data. This multifaceted analysis strategy overcomes limitations of available methods and permits identification of covarying gene programs in local niches at single cell type resolution.

### NiCo enables accurate and scalable cell type annotation of imaging-based spatial transcriptomics data

Cell type annotation of imaging-based spatial transcriptomics data on limited subsets of genes is a challenging problem. Available computational methods addressing these tasks were primarily optimized for spot-deconvolution of sequencing-based spatial transcriptomics. Nonetheless, state-of-the-art methods such as cell2location[27], Tangram[28], TACCO[29], and uniPort[30], can be applied to imaging-based data that have undergone cell segmentation. In a rigorous benchmarking analysis, we conducted a comprehensive comparison of NiCo annotations against annotations generated by Tangram, uniPort, TACCO and cell2location on published datasets, comprising mouse

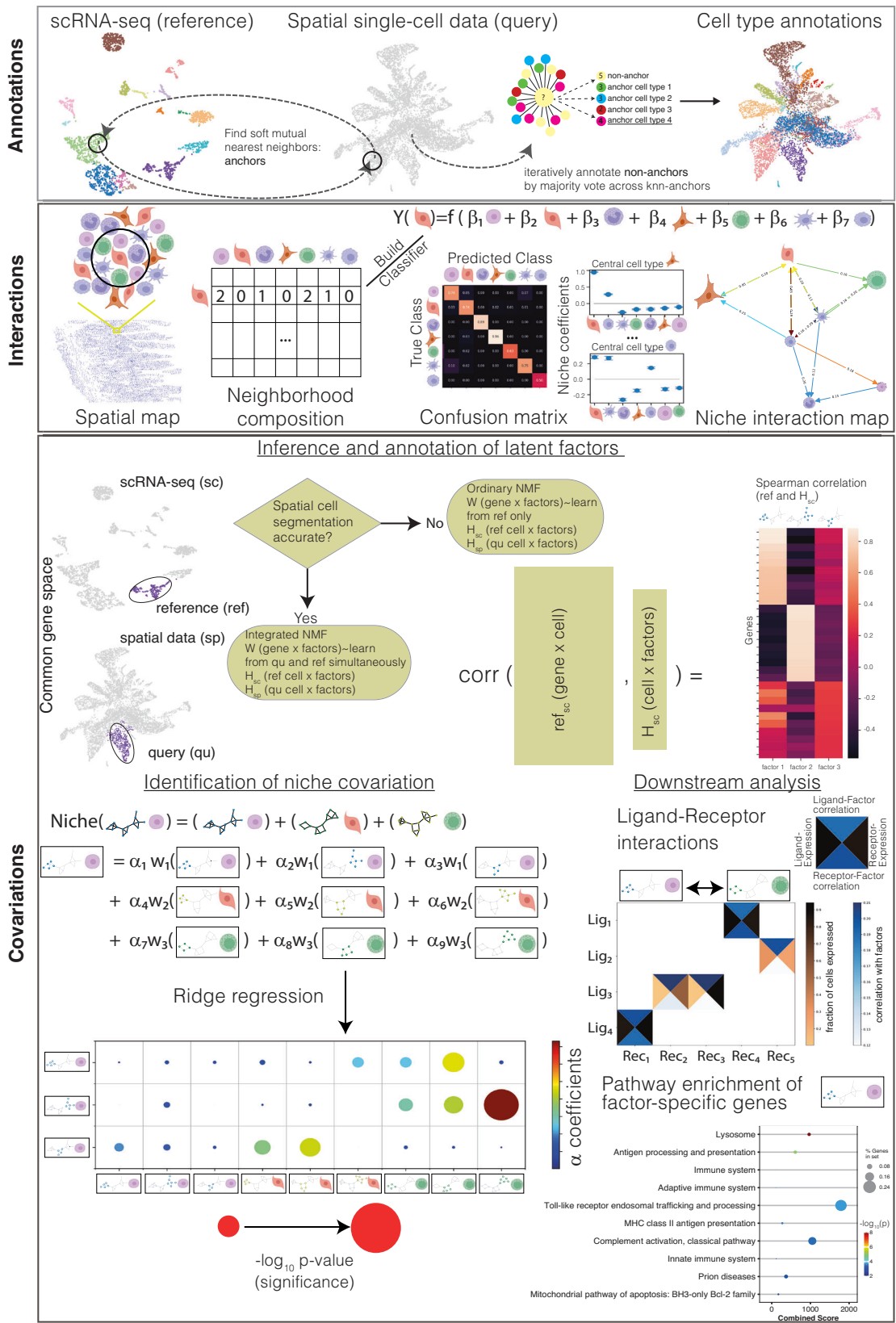

small intestinal MERFISH data[31], primary motor cortex MERFISH data[32], and mouse embryo seqFISH data[33] (Supplementary Fig. 1a). Considering the authors' annotation as ground truths, we evaluated the performance of these methods with well-established scoring metrics, i.e., adjusted Rand index and Jaccard similarity (Methods). Overall, NiCo outperformed Tangram and uniPort on all three datasets and

exhibited higher ground truth consistency on Primary Motor cortex and the embryo data than TACCO (Fig. 2a). On the intestinal data, NiCo yielded a higher Jaccard similarity but a lower adjusted Rand index than TACCO. Cell2location exhibited higher overlap with the ground truths on the intestinal dataset, and showed comparable performance on the Primary Motor Cortex data, while NiCo was more consistent

**Fig. 1 | Exploring cell state covariation in local niches with NiCo.** The schematic illustrates the three modules of the NiCo pipeline. *Annotations*: query imaging-based spatial transcriptomics data are annotated by label transfer from reference scRNA-seq data using a soft mutual nearest neighbor approach to derive anchors followed by iterative annotation of non-anchors. *Interactions*: For each cell type a logistic regression classifier is trained to predict the cell type identity from the niche composition. Predicted coefficients for each cell type are informative on predictive cell types within the niche and a cell type neighborhood graph is derived from the regression coefficients. *Covariations*: First, latent variables are inferred for each cell type to capture cell state variability using non-negative matrix factorization (NMF). The gene-by-factor matrix is learned simultaneously from reference

and query data; if the cell segmentation is imperfect as indicated, e.g., by "spill over" of cell type-specific markers, it can be learned only from reference data and transferred to the spatial modality. Factors can be associated with full transcriptome information based on gene-factor correlations derived from the scRNA-seq data. A ridge regression infers the dependence of each factor of the central cell type on all factors of the niche-cell types. Significance and magnitude of the regression coefficients indicating factor covariation in co-localized cell types, can be inspected in a dot plot. Covarying factors are interrogated for enriched pathways and ligand-receptor pairs to functionally interpret niche interactions. Created in BioRender. Grün, D. (2024) https://BioRender.com/v74f094.

with the ground truths for the embryo data (Fig. 2a). However, on large datasets such as the MERSCOPE liver dataset with ~400k cells (Data Availability), cell2location took more than two days to run, while NiCo finished in less than three hours on an AMD Ryzen 9, 5950X, 16-core processor with 128 GiB memory (Fig. 2b). Therefore, NiCo enables scalable reference-based cell type annotation of imaging-based spatial transcriptomics data on a standard laptop.

## NiCo identifies predictive niche interactions

To test NiCo's ability to predict niche interactions, we simulated spatial architectures of tissues with six cell types. To simulate preferential niche interactions between these cell types, we modeled mutual affinities by specifying Lennard-Jones potentials of varying attractive force ε between specific pairs of cell types (Methods). The simulated particle configurations thus reflect preferential niche interactions, which can be ranked by the strengths of their relative attractive forces. We assessed NiCo's regularized logistic regression classifier on these simulated data. Inspection of the confusion matrix, comparing predicted and ground truths cell types for two scenarios, demonstrates NiCo's capacity to predict cell type identities based on niche composition as a function of the simulated interaction strength (Fig. 2c–e). Magnitude and signs of the regression coefficients were consistent with the simulated interactions, confirming NiCo's capability for niche prediction. In the first scenario (Fig. 2c), NiCo predicted the strongest interaction for the central cell type T3 with T5 ($\varepsilon = 5$), and for central cell type T0, NiCo predicted the strongest interaction with T2 ($\varepsilon = 3$). Testing a more complex configuration (Fig. 2d), NiCo predicted the strongest interaction for the central cell type T3 with T2 ($\varepsilon = 10$), the second-strongest interaction with T1 ($\varepsilon = 8$), and the third-strongest with T5 ($\varepsilon = 5$). Similar conclusions hold true for the other cell types. For a larger neighborhood radius ($R = 5$, Fig. 2e), the relative rankings are not always mirrored by the inferred predictors due to stronger second-order effects. In general, we recommend running NiCo with radius $R = 0$ to focus on juxtacrine interactions before exploring larger radii. MISTy[22] is an available machine learning framework for predicting structural and functional interactions from spatial omics data. To benchmark with NiCo's interaction module, we applied MISTy to the second simulated scenario for juxtacrine interactions (Fig. 2d, Methods). MISTy recovered the relative ranking by interaction strength with some exceptions (Fig. 2f). For instance, the best predictor of T5 were T0 and T4 instead of T3. However, since MISTy interaction coefficients reflect predictor importance derived from random forests regression, a high value could indicate positive or negative interactions, which makes a direct interpretation difficult. Similarly, MISTy only picks up T3 as predictor of T2, but fails to detect T0. Moreover, T5 was not inferred as predictor of T3. The predictive contribution of higher-order interactions becomes apparent when inspecting MISTy's paraview and comparing to NiCo with radius $R = 5$ (Fig. 2e), e.g., for predicting T5: both methods now detect T4 and T2 as best predictors. However, NiCo reveals that T4 is depleted while T2 is enriched in the paraview. Overall, NiCo's predictions were more consistent with the ranking by simulated interaction strengths.

## Predicted niche interactions are consistent with tissue domain architecture

A key objective of spatial transcriptomics analysis is the identification of topological tissue domains with distinct cell type architectures. Although NiCo was not designed specifically for this task, its predicted cell type interactions are reflective of tissue domain architecture. To benchmark NiCo with tissue domain detection methods, including Stagate[34], Seurat BuildNicheAssay[35], CellCharter[36], SpaGCN[37], SpatialPCA[38], and Banksy[39] we ran each method on spatial MERFISH whole mouse brain data with available ground truths annotation of six regional tissue domains comprising 17 cell types in total (Methods)[40]. For each domain, we selected the NiCo cell type with the highest correlation of the regression coefficients to the domain-specific cell type frequencies, and compared to the cell type enrichment computed for the domains predicted by each method (Supplementary Fig. 1b, c and 2). Overall, the interactions predicted by NiCo were on average more consistent with ground truths domain annotations than the domain predictions obtained from available methods (Fig. 2g). On another dataset of mouse visual cortex STARmap data with ground truths annotation of neocortical layers[18], NiCo's performance was comparable to the domain detection methods (Fig. 2h and Supplementary Fig. 3). We note that NiCo predicts global interaction coefficients as a basis for inferring cell state dependencies between interacting cell types, while domain detection methods predict local tissue regions with unique cell type compositions. The observation that NiCo interaction coefficients of specific cell types frequently reflect domain architecture indicates that these cell types dominate particular domains. Benchmarking of memory usage and computation time for niche domain detection on an AMD Ryzen 9, 5950X, 16-core processor (Fig. 2b) suggests that NiCo is both fast and memory-efficient compared to available niche detection methods and does not require any pre-specified number of clusters or cluster resolution parameters.

## NiCo deciphers cellular crosstalk during mouse organogenesis

In the following, we demonstrate the predictive capacity of NiCo on three different tissue environments with varying complexities and distinct architectural features. As a first scenario, we interrogate niche architecture and covariation of gene programs between local neighbors in the context of mouse embryonic development. We applied NiCo to seqFISH data of E8.5 embryos[33], where a diversity of organ tissues has already developed, and utilized an scRNA-seq reference data of the same developmental stage[41]. To facilitate data integration of the two modalities, we annotated cell types within the seqFISH dataset using the NiCo annotation module with default parameters to transfer the cell type labels from the scRNA-seq clusters (Fig. 3a, b).

With the emergence of organ structures and the establishment of three-dimensional spatial tissue domains, embryonic tissues pose a unique challenge when deciphering niche interactions[42,43]. We applied the interaction module of NiCo to interrogate the dynamic interplay within and across these tissue domains. The level of confusion of NiCo's cell type classifier reflects the predictive capacity of the niche

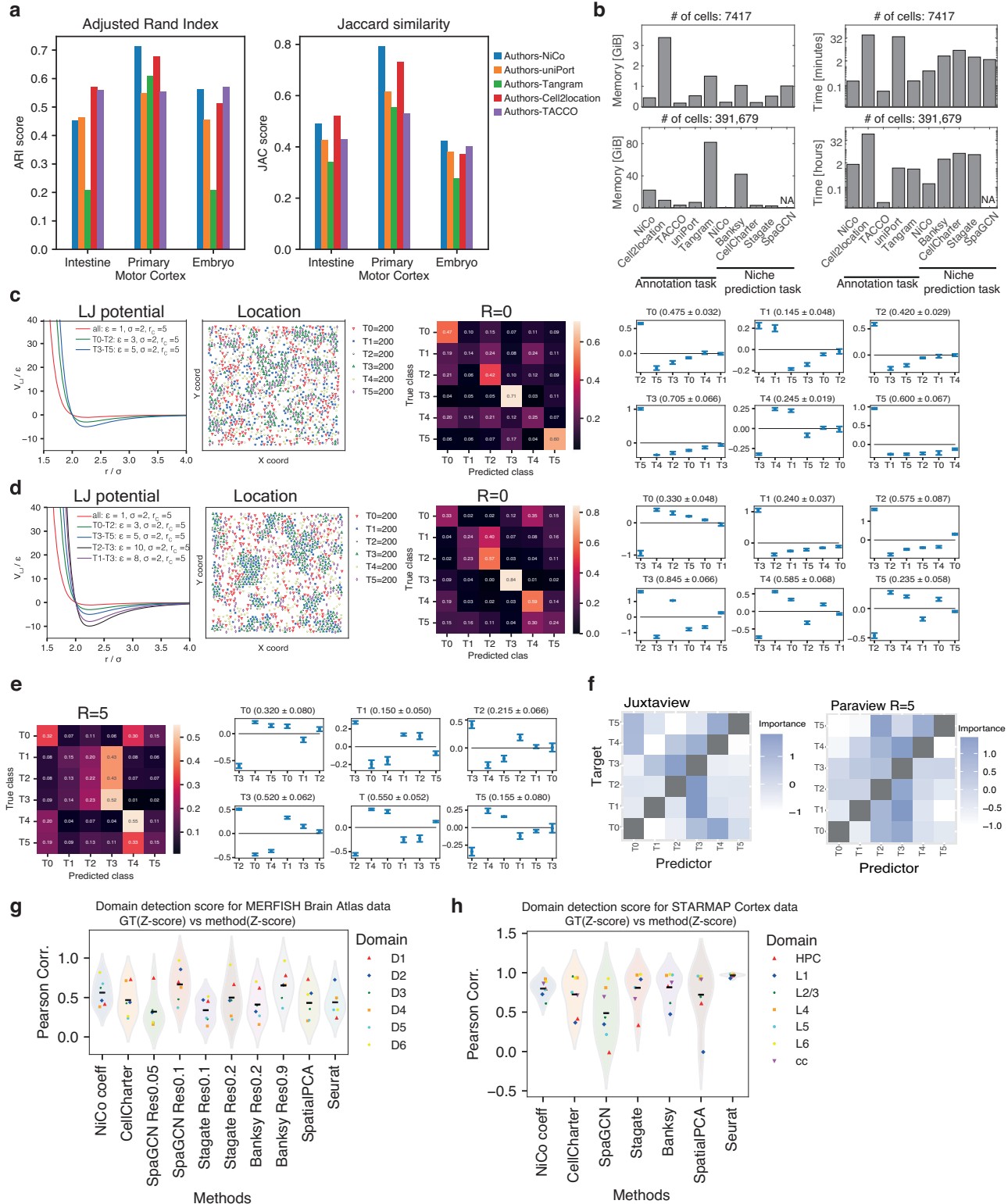

composition for each cell type. The majority of cell types in the E8.5 embryo were predicted well from their niche composition (Fig. 3c) with a weighted average precision of 0.71. This observation is not unexpected given the topological ordering of many cell types into tissue domains (Fig. 3a), e.g., gut, forebrain/midbrain/hindbrain, and heart tissue (cardiomyocytes). In these cases, the relative magnitude of regression coefficients suggests that the highest predictive capacity is contributed by neighbors of the same cell type identity (Fig. 3d). Conversely, elevated values in the off-diagonal entries of the confusion

matrix reflect similarities in the niche composition of distinct cell types, e.g., for endothelium and haematoendothelial progenitors (Fig. 3c). Hence, inspection of the confusion matrix enables delineation of cell types with highly specific niche composition, and pairs or even larger groups of cell types with similar niche composition.

The regression coefficients of each cell type can be utilized to construct a spatial cell type neighborhood graph highlighting dominant niche interactions. The sparsity of this graph can be controlled by a cutoff on the regression coefficients. After applying a cutoff of 0.01,

**Fig. 2 | Benchmarking NiCo annotation and interaction inference.**
**a** Benchmarking of NiCo annotations with uniPort, Tangram, TACCO, and cell2location for published datasets (mouse intestine and primary motor cortex MERFISH data, and mouse embryo seqFISH data; see text for details) with available ground truths annotations (authors' annotation). Consistency of annotations between two methods was evaluated by Jaccard similarity (JAC) and adjusted R and index (ARI). **b** Memory requirement (left) and run time (right) for the cell type annotation and niche prediction task on intestinal MERFISH data (7,417 cells, top) and liver MER-SCOPE data (391,679 cells, bottom). SpaGCN was aborted on the liver datasets due to exhausted memory. Runtime is shown on a log$_2$ scale. **c**, **d** Simulation experiments for testing NiCo's interaction module. The locations of cells were simulated for six cell types in two dimensions (2D) using Lennard-Jones (LJ) pairwise interaction potentials (first panel). Simulated cell type locations were used as input to NiCo's interaction module (second panel). The confusion matrix (third panel) highlights the predictive capacity of the neighborhood composition. Classifier coefficients (fourth panel) are ordered by magnitude, reflecting co-localization preference, or interaction strength. The confidence score is indicated in parentheses. A more uniform (**c**) and a more biased (**d**) scenario were simulated and assessed based on direct neighbors. See Methods for explanations of the parameters. **e** Same as (**d**) but using a larger neighborhood radius (R = 5). (c-e) error bars on β coefficients indicate standard deviation derived from five-fold cross-validation (Methods). **f** Niche cell type interactions inferred with MISTy for direct neighborhoods (left: juxtacrine view, right: paraview radius 5) on the more complex simulation scenario (**d**). The two-dimensional map indicates the importance of neighboring cell types (Predictor) for predicting central cell types (Target). **g**, **h** Comparison of cell type interactions predicted by NiCo with cell type enrichment in topological tissue domains predicted by tissue domain detection methods (CellCharter, SpaGCN, Stagate, Banksy, SpatialPCA, Seurat) on (**g**) Allen brain MERFISH atlas data and (**h**) STARmap visual cortex data. The Pearson correlation between predicted cell type enrichment Z-score and cell type enrichment in best matching ground truths (GT) tissue domains is compared (Methods). For NiCo, the cell type with the highest correlation of its regression coefficients to the Z-score was selected. Source data are provided as Source Data file (Data Availability).

the neighborhood graph (Fig. 3e) unveils well known tissue co-localization patterns, such as *extraembryonic (ExE) endoderm−definitive endoderm − gut*, reflecting the endodermal origin of gut epithelium, or *endothelium − haematoendothelium − blood progenitors* (Fig. 3d), consistent with the emergence of primitive blood cells from yolk sac endothelium at this stage. The high connectivity of blood progenitors and erythroid cells is likely due to presence of these cells within the vasculature.

NiCo's covariation module identifies covarying molecular programs in co-localized cell types encoded by cell type-specific latent variables, or factors. To explore covariation between co-localized cell types in common niches, we inferred three latent factors for each cell type, capturing major intra-cell-type variability (Methods). Using NiCo's ridge regression step, we detected covariation between these factors across pairs of co-localized cells.

Focusing on cardiac tissue, this analysis revealed strong positive covariation of each cardiomyocyte factor with the same factor in neighboring cardiomyocytes, suggesting cell state synchronization in neighboring cardiomyocytes (Fig. 3f). Another significant covariation ($\log_{10} P = -3.92$) was detected between cardiomyocyte factor (Fa) 2 and pharyngeal mesoderm Fa1, supported by 223 co-localized pairs out of 853 pharyngeal mesoderm cells and 773 cardiomyocytes (Fig. 3f). During mouse embryonic development, the vital role of the pharyngeal mesoderm is well documented, contributing significantly to extensive regions of both head and heart muscle[44]. Heart formation involves two distinct cardiac progenitor cell populations: the first heart field (FHF) and the second heart field (SHF)[45]. The T-box transcription factor Tbx5 marks the FHF, which is responsible for the left ventricle and atria development[46]. Conversely, around E8.0, the SHF, originating from the pharyngeal mesoderm and characterized by Islet1 (Isl1) expression, populates the cardiac outflow tract and right ventricle[47,48]. Studies suggest that canonical and noncanonical WNT signaling are essential for progenitor cell proliferation in pharyngeal mesoderm and the development of SHF heart differentiation[49]. Anomalies in the signaling communication between the pharyngeal mesoderm and cardiomyocytes are associated with congenital heart diseases[50].

To interpret this covariation pattern, we inspected the top positively and negatively correlating genes with cardiomyocyte Fa2 and pharyngeal mesoderm Fa1, computed from the scRNA-seq reference data to leverage genome-wide information (Fig. 3g). While key markers for cardiac progenitors such as *Nkx2-5*, *Isl1*, *Mef2c*, and *Fgf8* were negatively correlated with pharyngeal mesoderm Fa1, mature cardiac genes involved in muscle contraction, e.g., cardiac troponins (*Tnnt2*, *Tnni3*, *Tnnc1*, *Tnnt1*, *Tnni1*), and other mature cardiomyocyte markers such as *Myl2*, *Myl3* and *Acta* anticorrelated with cardiomyocyte Fa2. These associations were also reflected by enriched biological processes among the top positively and negatively correlating genes

which further highlights an association of pharyngeal mesoderm Fa1 with translational activity (Supplementary Fig. 4a, b). To explore signaling interactions that may mediate this interaction, we inspected ligands and receptors correlated to the covarying factors, and identified a potential interaction between *Wnt2* on cardiomyocytes and *Fzd2* on pharyngeal mesodermal cells (Fig. 3h), consistent with the known requirement of WNT signaling for SHF progenitor proliferation and differentiation from pharyngeal mesoderm[49]. Moreover, the secreted signaling mediator *Hmgb1* is among the top three genes most highly correlated to Pharyngeal mesoderm Fa1. Early knockout of this gene in the developing hearts leads to cardiac growth retardation[51]. Therefore, Hmgb1 signaling from the pharyngeal mesoderm could be an inducer of cardiomyocyte maturation (Fig. 3g). These observations indicate the co-localization of immature cardiomyocytes with translationally active putative cardiac progenitors in the pharyngeal mesoderm, consistent with the SHF developmental origin, highlighting the capability of NiCo to detect cell state covariation in local niches underlying embryonic tissue development.

## NiCo unveils covariation of intestinal stem cell and Paneth cell states

The intestinal epithelium undergoes permanent homeostatic turnover fueled by an active stem cell compartment. This renewal process maintains the relative proportions of various cell types along the crypt-villus axis. Lgr5+ stem cells, localized at the bottom of the intestinal crypts, give rise to proliferating transient-amplifying (TA) progenitors and eventually differentiate into mature epithelial cell types[52]. This differentiation process is orchestrated with an upward migration from the crypt bottom to the tip of the villus. Recent observations indicate that Paneth cells, a known stem cell niche, could differentiate directly from stem cells[53].

To conduct a spatially resolved analysis of intercellular crosstalk within the mouse small intestine, we applied NiCo to published MER-FISH data[31]. Integration with a comprehensive scRNA-seq dataset[54] facilitated cell type annotation in the spatial domain reflecting known localization of cell types along the crypt-villus axis (Fig. 4a, b). While stem and TA cells (stem/TA) as well as Paneth cells were localized to the crypt bottom, enterocytes segregated into bottom, medium, and top zone enterocytes. Goblet, tuft, and enteroendocrine cells were distributed more randomly along the crypt villus axis. NiCo's interaction analysis revealed a predictive capacity of the niche composition for the majority of cell types (Fig. 4c). Off-diagonal elements in the confusion matrix, indicative of a similar niche composition for the respective cell types, suggest that the goblet cells share a similar niche with the stem/TA cells and bottom zone enterocytes, in line with their preferential localization to the lower crypt region. Inspection of the regression coefficients indicated that stem/TA cells were predicted by

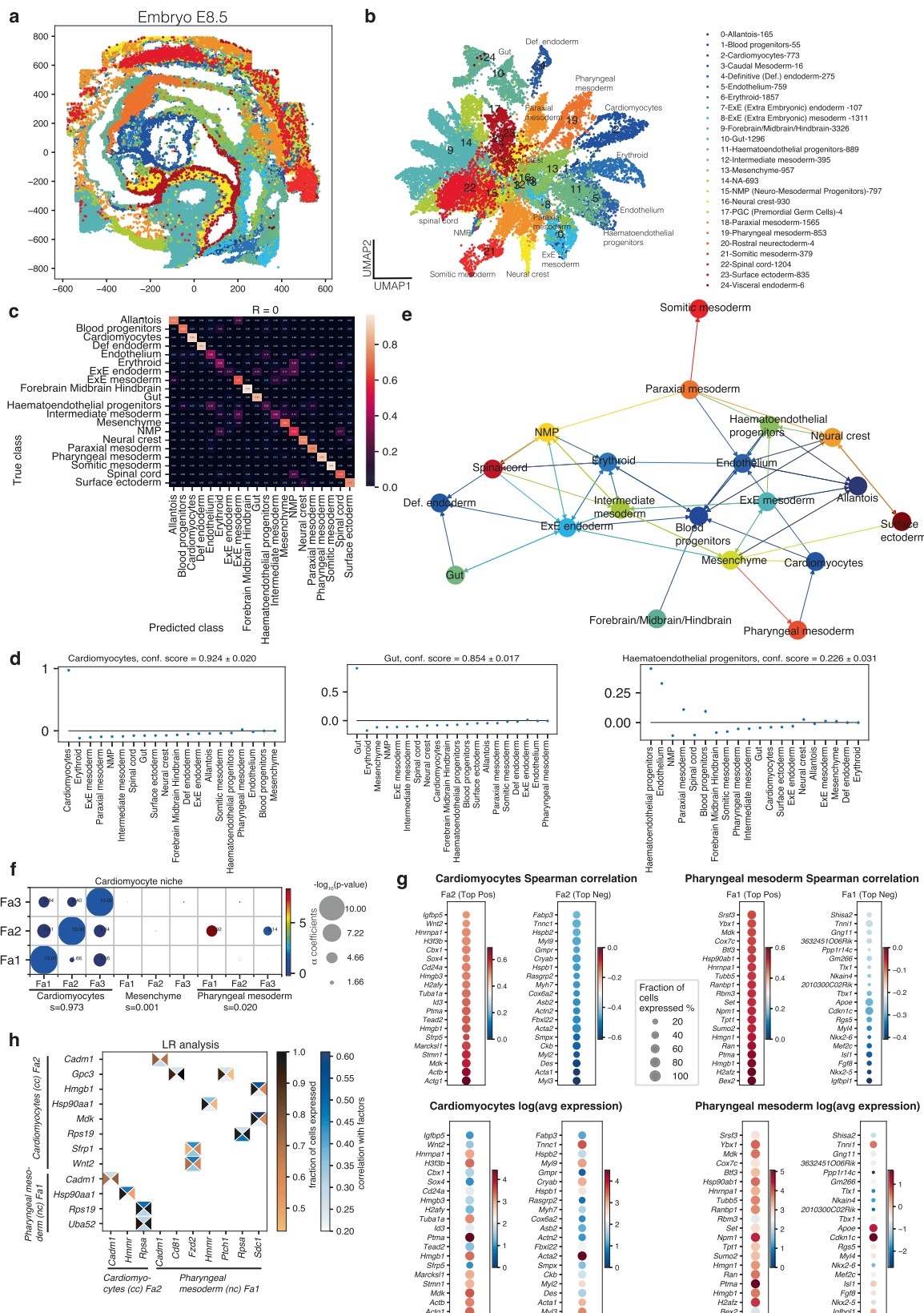

the presence of goblet and Paneth cells within their niche, and, vice versa, Paneth cells were predicted by co-localized Paneth and stem/TA cells (Fig. 4d). The cell type neighborhood graph derived from the regression coefficients segregated into a module of epithelial cells, and a module connecting different types of immune cells, lymphatic endothelial cells, as well as vascular cells (Fig. 4e).

Focusing on the stem/TA cell niche, NiCo's covariation module inferred a significant covariation of stem/TA cell Fa1 with goblet cell Fa2 ($\log_{10}P = -4.00$) and Paneth cell Fa1 ($\log_{10}P = -1.43$) supported by 217 co-localized pairs out of 1107 stem/TA cells and 184 Paneth cells (Fig. 4f). Inspection of the top-correlating genes with stem/TA Fa1 in the scRNA-seq data revealed several known stem cell markers such as

**Fig. 3 | NiCo identifies niche architecture and covariation during mouse embryogenesis. a** Spatial representation of cell type annotations obtained with NiCo. See legend in (**b**). **b** UMAP representation based on gene expression of the spatial data. Cell type annotations obtained with NiCo are highlighted. The legend indicates the number of cells for each cell type. **c** Confusion matrix highlighting predictive capacity of the niche for all cell types. **d** Regression coefficients reflecting local niche interactions for cardiomyocytes, gut, and haematoendothelial progenitors. Regression coefficients (*y*-axis) were ordered by decreasing magnitude (*x*-axis). Error bars, see "Methods". **e** Cell type interaction map derived from the regression coefficients of NiCo's interaction module (Methods). A cutoff of c = 0.01 was applied. Arrows point from predictive niche cell types towards the central cell type. **f** Covariation between cardiomyocyte factors (*y*-axis) and co-localized neighborhood cell type factors (*x*-axis). Circle size scales linearly with -$\log_{10}$(*p*-value), and circle color indicates ridge regression coefficients. S denotes the normalized niche coefficient score. The multivariate regression *p*-value was derived from two-tailed *t*-statistics. **g** Spearman correlations (top) and average expression in the corresponding cell type (bottom) for the top 20 positively and negatively correlated genes from scRNA-seq reference data for pharyngeal meso-derm factor (Fa) 1 and cardiomyocyte Fa2. **h** Ligand-receptor pairs correlated with pharyngeal mesoderm Fa1 and cardiomyocyte Fa2 (cc, central cell; nc, niche cell). The rectangle's north and south faces represent ligand and receptor correlation to the factors, while west and east faces represent the proportion of ligand and receptor expressing cells. Ligands, *y*-axis; receptors, *x*-axis. See "Methods".

*Olfm4*, *Lgr5*, and *Hopx*. Paneth cell Fa1, on the other hand, strongly correlated with markers of Paneth cell progenitors (Fig. 4g and Sup-plementaryFig. 5a, b). This pattern suggests that the stemness sig-nature covaries with a progenitor signature in neighboring Paneth cells. We note that Paneth cells can emerge directly from stem cells and the nascent Paneth cell progenitors may thus signal to the sister stem cells in order to maintain their stemness[53] (Supplementary Fig. 5c, d). Interrogation of ligand-receptor pairs correlated to the covarying factors recovered the *Wnt3* ligand correlating to Paneth cell Fa1 and *Fzd2* as well as *Fzd7* receptors correlating to stem/TA cell Fa1 (Fig. 4h). Canonical Wnt signaling through the Wnt3-Fzd7 axis is essential for the survival of Lgr5+ intestinal stem cells[55]. Reassuringly, regulation of canonical Wnt signaling as well as negative regulation of development were identified among the enriched gene ontology (GO) biological processes enriched in top correlating genes of stem/TA Fa1 (Supple-mentary Fig. 5e). Our spatial covariation analysis extends previous findings by identifying Paneth cell progenitors that likely directly emerged from the stem cell pool as the source of the critical Wnt ligand required for maintenance of the stem cell[53]. Consistently, *Wnt3* is downregulated during Paneth cell maturation following the trend of Fa1 (Fig. 4i). We note that existing methods for the inference of spatial cell-cell crosstalk, such as Niche-DE[56], COMMOT[21], stLearn [57]and CellNeighborEX [58]did not recover this interaction due to their limita-tion to directly measured gene sets (Methods and Supplementary Fig. 6). In summary, NiCo recovers the critical Paneth cell niche of intestinal stem cell maintenance without any prior information and predicts nascent Paneth cell progenitors as a critical *Wnt* ligand source.

## NiCo predicts covariation of stellate cell and Kupffer cell states in the mouse liver

The liver is the major metabolic organ in the body and possesses a remarkable regenerative capacity[59]. Liver tissue is functionally orga-nized into hexagonally shaped units termed lobules[60]. Within each lobule, hepatocytes are organized into distinct zones along the axis connecting the central vein in the center with the portal vessels at the boundary. Metabolic and signaling pathways exhibit zonal gradients, a phenomenon also reflected by transcriptome zonation in hepatocytes and sinusoids connecting the central vein with the portal vein and the hepatic artery[61,62]. Achieving spatial single-cell resolution is imperative to reveal the intricate cellular interactions within zonated liver niches. We applied NiCo to publicly available highly-multiplexed smFISH data on 347 genes in the mouse liver generated with the MERSCOPE plat-form (Data Availability) and integrated these data with a comprehen-sive single-cell/single-nucleus RNA-seq mouse liver atlas[63].

NiCo could annotate 375,161 out of 391,679 cells (95.8%) in the spatial modality (Fig. 5a). Since no author annotation was available as ground truths, we inspected spatial cell type localization within the lobule. According to NiCo's annotation, portal vein endothelial cells localized to the margin of tissue areas that morphologically resemble large blood vessels, distant from smaller areas demarcated by the presence of central vein endothelial cells. Reassuringly, portal hepa-tocytes were annotated in vicinity to portal vein endothelial cells, while central hepatocytes co-localized with central vein endothelial cells, and the intermediate zone was populated by mid-zonal hepatocytes (Fig. 5b, c). This observation demonstrates NiCo's capacity to correctly annotate cell states with more subtle transcriptional differences such as hepatocytes within different zones. A comparison with alternative state-of-the-art annotation tools such as cell2location, Tangram, TACCO and uniPort revealed that these methods struggle with the recovery of zonated hepatocyte states (SupplementaryFig. 7). Apart from NiCo, TACCO and cell2location managed to annotate hepato-cytes of different zones correctly, although the separation of mid-zonal and portal hepatocytes was less clear. Moreover, on this dataset it took more than two days of computation time on a standard machine to run cell2location, while NiCo finished in few hours (Fig. 2b).

The logistic regression classifier of NiCo's interaction module identified cell types with a highly predictive niche, such as portal and central vein endothelial cells, or cholangiocytes (Fig. 5d), and the regression coefficients confirmed known associations such as co-localization of central hepatocytes and central vein endothelial cells (Fig. 5e). Various immune cell populations, on the other hand, such as dendritic cells, monocytes, or neutrophils, were not predicted by the cell type composition of their niche suggesting a variable neighbor-hood for these migratory cells. The cell type neighborhood graph derived from the regression coefficients segregated into a portal cell type module, comprising portal vein and lymphatic endothelial cells, as well as cholangiocytes and portal hepatocytes, and a mid-central module of central and mid-central hepatocytes, central vein and sinusoidal endothelial cells (Fig. 5f). Both modules were connected to fibroblasts, which are known to reside around the central and portal blood vessels.

To understand whether the niche interactions identified from the regression coefficients were meaningful even in situations where the predictive capacity was reduced, we focused on the stellate cell niche, which exhibited a classification accuracy of 0.06. Nonetheless, the logistic regression classifier suggested specific positive associations with sinusoidal endothelial cells, Kupffer cells, and central/mid-zonal hepatocytes (Fig. 5e). This niche composition aligns perfectly with a recent characterization of the stellate cell niche[64]. We demonstrate robustness of this niche prediction across technical replicates. Run-ning NiCo on a second liver slice analyzed with MERSCOPE recovers liver cell type interactions in general, and the stellate cell-Kupffer cell niche composition in particular (Supplementary Fig. 8). Moreover, to test the robustness of NiCo to sample size, we repeated the analysis on a smaller sub-region containing only 88,772 (22.7%) cells. Again, NiCo recovered liver cell type interactions and the stellate-Kupffer cell niche (Supplementary Fig. 9). To explore the consequence of these niche interactions for the state of stellate cells, we applied NiCo's covariation module to the full slice data. The most significant covariation ($\log_{10}P = -3.08$) was inferred between stellate cell Fa2 and Kupffer cell Fa1 (Fig. 6a), supported by 7781 co-localized pairs out of 11,881 stellate cells and 38,932 Kupffer cells. Stellate cells are quiescent in the normal liver, but undergo activation and transdifferentiate into myofibro-blasts producing alpha-smooth muscle actin (SMA) and collagen after

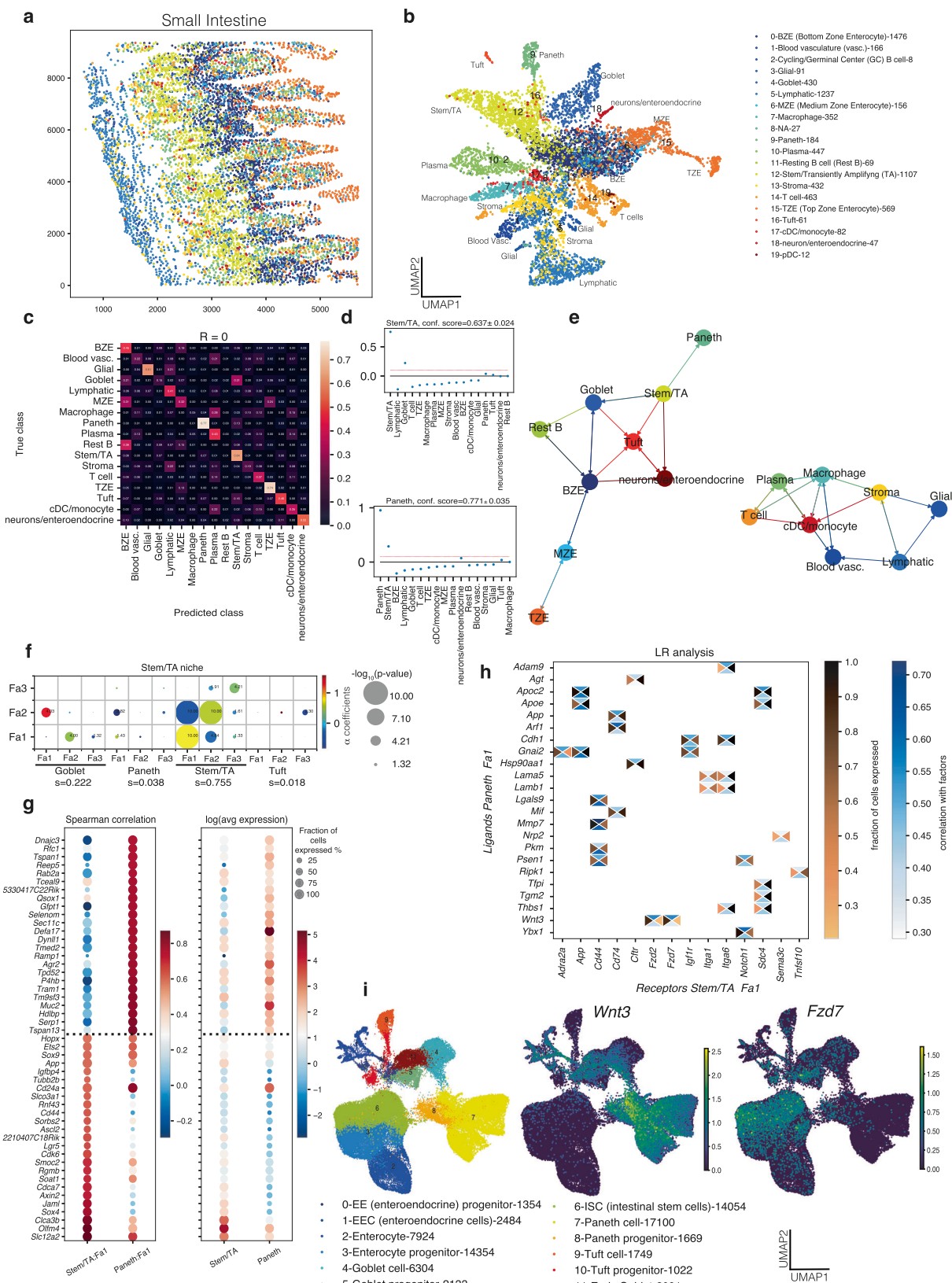

chronic liver damage[65,66]. Kupffer cells function as resident macrophages and shield the liver from bacterial infections[67]. The top correlating genes with stellate cells Fa2 (Fig. 6b) were enriched in viral defense and antigen presentation pathways (Supplementary Fig. 10a, b), indicative of a proposed function as innate immune cell[68]. On the other hand, proteoglycans such as *Dcn* and *Bgn* were among the

top correlating genes. Antigen presentation and complement activation pathways were highly enriched among the top correlating genes with Kupffer cell Fa1 reflecting an activated Kupffer cell state (Fig. 6b and Supplementary Fig. 10a–e). Focusing on ligand-receptor pairs correlated with the covarying factors, we detected the ligand *Tgfb1* on Kupffer cells and the receptor *Tgfbr3* on stellate cells (Fig. 6c).

**Fig. 4 | NiCo infers cell state covariation between mouse intestinal stem cells and Paneth cell progenitors. a** Spatial representation of cell type annotations obtained with NiCo. See legend in (**b**). **b** UMAP representation based on gene expression of the spatial data. Cell type annotations obtained with NiCo are highlighted. The legend indicates the number of cells for each cell type. cDC, conventional dendritic cell; pDC, plasmacytoid dendritic cell. **c** Confusion matrix highlighting predictive capacity of the niche for all cell types. **d** Regression coefficients reflecting local niche interactions for stem/TA cells (top) and Paneth cells (bottom). Regression coefficients (*y*-axis) were ordered by decreasing magnitude (*x*-axis). The red dotted line indicates the threshold for cell type interaction (c = 0.1) applied in (**e**). Error bars, see "Methods". **e** Cell type interaction map derived from the regression coefficients of NiCo's interaction module (Methods). Arrows point from predictive niche cell types towards the central cell type. **f**, Covariation between stem/TA factors (*y*-axis) and co-localized neighborhood cell type factors (x-axis). Circle size scales linearly with -log$_{10}$ (*p*-value), and circle color indicates ridge regression coefficients. S denotes the normalized niche coefficient score. The multivariate regression *p*-value was derived from two-tailed t-statistics. **g** Spearman correlations (left) and average expression (right) for the top 20 positively correlated genes from scRNA-seq reference data for stem/TA Fa1 and Paneth Fa1. Dashed line demarcates the genes associated with each cell type. **h** Ligand-receptor pairs correlated with stem/TA cell Fa1 and Paneth cell Fa1 (cc, central cell; nc, niche cell). The rectangle's north and south faces represent ligand and receptor correlation to the factors, while west and east faces represent the proportion of ligand and receptor expressing cells. Ligands, *y*-axis; receptors, *x*-axis. See "Methods". **i** UMAP visualization of intestinal cell types (left) and normalized expression of *Wnt3* (center) and *Fzd7* (right). Data from Böttcher et al.[53]. Cell counts are indicated in the legend.

Tgf-β acts as a potent inducer of fibrosis in the liver[69,70]. Proteoglycans such as *Dcn* and *Bgn* are produced by stellate cells and exert known anti-fibrogenic functions; they inhibit Tgf-β signaling by sequestering Tgf-β ligands at the extracellular matrix or cell surface[71]. The predicted correlation of the *Tgfb1*-*Tgfbr3* interaction with antigen presentation and immune activation in Kupffer cells and similar functions as well as proteoglycan production in co-localized stellate cells (Fig. 6d, e and Supplementary Fig. 10c–e), may indicate a functional dependence of these processes in the healthy liver. Sensing of low levels of viruses and other pathogens may lead to activation of Kupffer cells and induction of Tgf-β signaling, which is received by interacting stellate cells. In stellate cells, this signal could induce proteoglycans which sequester Tgf-β ligands to dampen pro-fibrogenic signaling in order to avoid stellate cell activation at low pathogen levels in the healthy liver. Upon chronic stimulation, this mechanism may be insufficient to inhibit Tgf-β signaling surpassing a critical threshold, and thus leading to full stellate cells activation and collagen production.

Since *Tgfb1* and *Dcn* were not directly measured in the original MERSCOPE experiment, we independently validated the inferred covariation between *Tgfb1* in Kupffer cells marked by *Clec4f* and *Dcn* in stellate cells by single-molecule hybridization chain reaction (smHCR)[72] (Methods). Indeed, this highly sensitive assay confirmed a strong correlation (Pearson's correlation coefficient r = 0.59) between *Tgfb1* and *Dcn* in co-localized pairs of stellate and Kupffer cells within healthy liver tissue (Fig. 6f, g and Methods). In comparison, correlation of *Dcn* in stellate cells with the general marker *Clec4f* in co-localized Kupffer cells was markedly reduced (Pearson's correlation coefficient r = 0.24) indicating the specificity of the covariation between *Dcn* and *Tgfb1* in pairs of stellate and Kupffer cells (Fig. 6g). We finally validated dampening of hepatic stellate cell (HSC) activation by Dcn in vitro (Methods). qPCR quantification after in vitro culture of HSCs stimulated with Tgf-b in the presence (HSC+Tgfb +Dcn) or absence (HSC+Tgfb) of Dcn suggested dampened activation when Dcn was administered to the culture. Specifically, HSC stimulation by Tgf-b in the presence of Dcn led to significantly lower induction of activation markers *Col1a1* (*P* < 0.01) and *Pdgfrb* (*P* < 0.04), while the quiescent HSC marker gene *Reln* remained unaffected. (Methods, Fig. 6h and Supplementary Fig. 10f). Induction of activation markers *Lox* and *Acta2* was also reduced although the fold-reduction did not reach significance. These observations support the hypothesis that secreted Dcn may indeed dampen HSC activation by sequestering Tgf-b. The key insight infered by NiCo is the induction of *Dcn* expression in HSCs as a result of Tgf-b upregulation in co-localized Kupffer cells, supported by the validated covariation pattern (Fig. 6g). This example showcases the ability of NiCo to generate insightful hypotheses about the functional implications of cell type-specific niche interactions in the liver.

### NiCo detects covariation patterns in sequencing-based mouse cerebellum spatial transcriptomics data

Although NiCo was primarily designed for the analysis of single-cell resolution spatial transcriptome data, we evaluated its capabilities on a mouse cerebellum dataset generated with Slide-seqV2, a sequencing-based spatial transcriptomics method with 10 μm resolution[73]. Since pixels typically consist of 1-2 cells, interaction and covariation inference may be confounded to a certain extent. Utilizing cell type annotations for each pixel generated by the authors, we first ran NiCo's interaction module, which recovered well known interactions, e.g., between Purkinje neurons and a specialized form of astrocytes called Bergmann glial cells[74] (Supplementary Fig. 11). Covariation analysis between these co-localized cell types recovered covarying secretory programs, e.g., calcium signaling from Bergmann cells to Purkinje neurons[75] (Supplementary Fig. 11e). This vignette showcases the general applicability of NiCo to sequencing-based spatial transcriptomics.

## Discussion

With the availability of single-cell sequencing methods, our understanding of the molecular definition of cell types has substantially advanced. As a key insight, it was recognized that a cell type cannot be defined by a unique transcriptome state, but rather populates a continuous region within the high-dimensional transcriptome space, reflecting systematic cell state variability. Such variability could in theory be accounted for by cell-intrinsic factors driven by the architecture of the gene regulatory network governing a cell type, which could give rise to multi-stability due to stochasticity of molecular interactions[76]. However, cells within tissues are exposed to extrinsic inputs from the microenvironment, which could be another mechanism to induce cell state modulations. The ability to deconvolve intrinsic and extrinsic cell state determinants represents a critical step forward towards understanding robustness and plasticity of cell types in multicellular tissue environments.

Application of single-cell resolution spatial transcriptomics permits addressing this challenge. NiCo explains cell state variability by gene expression covariates within cell types populating the same niche. To avoid strong assumptions on the signaling mediators, we did not incorporate ligand-receptor interactions as a constraint. Hence, predicted covariations may result from direct signaling interactions but also from indirect sources such as cellular adaptation towards biomechanical cues, or competition for metabolites[77]. Nonetheless, NiCo proposes candidate signaling pathways mediating predicted covariations by identifying ligands and receptors correlating to the covarying factors within the interacting cell types. However, we believe that alternative drivers of cell state covariation in local niches are important to consider, and therefore provide a tool to detect the downstream effects on the transcriptional state of a cell unconstrained by assumptions on the mediators of this interaction.

Since imaging-based technologies typically measure few hundred genes, integration with scRNA-seq data is critical for the functional interpretation of predicted covariations between co-localized cells. NiCo solves this challenge by inferring a small set of latent factors capturing dominant cell state variability for each cell type, and utilizes these factors to infer cell-cell covariation and to

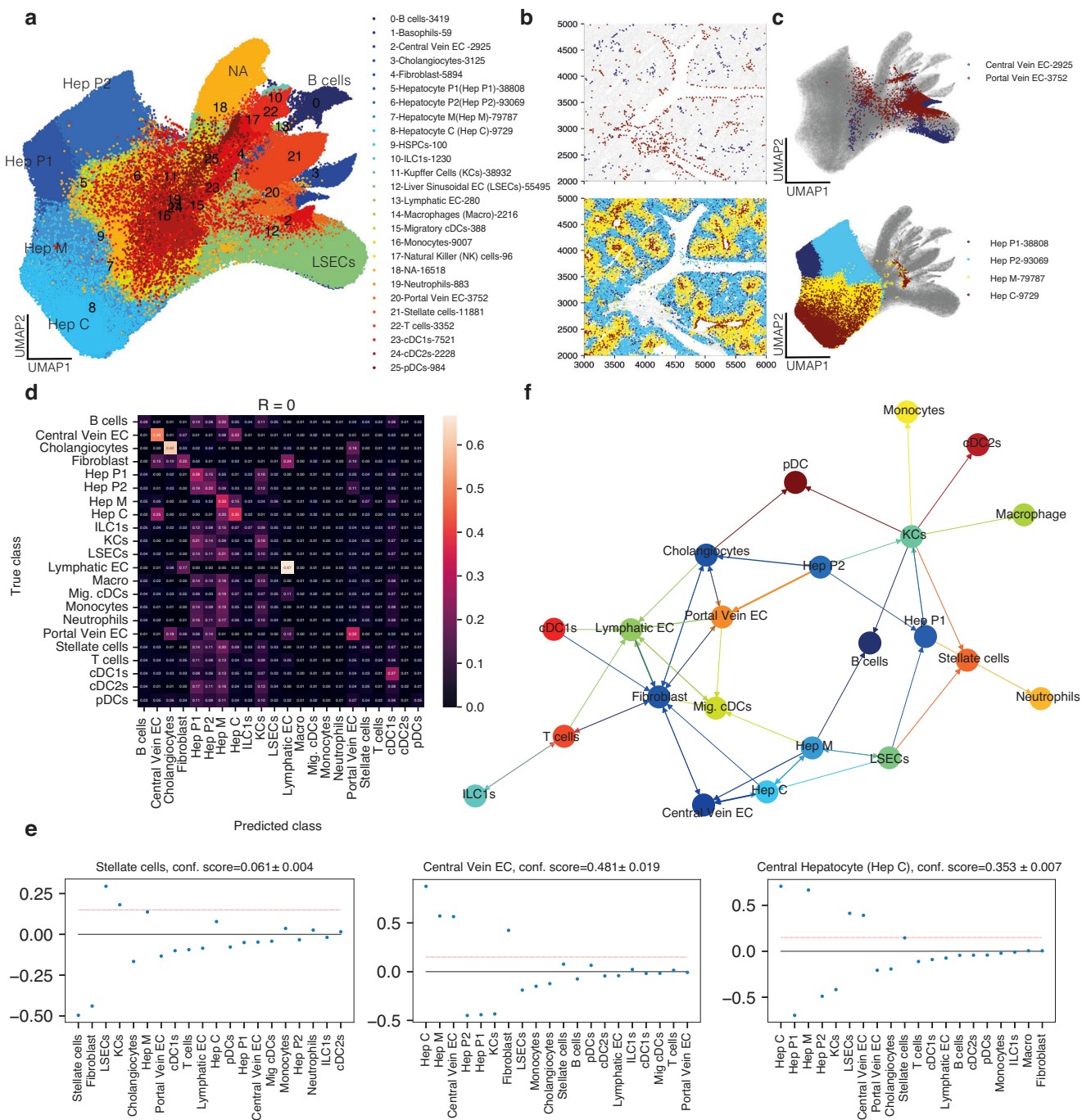

**Fig. 5 | NiCo recovers niche architecture of the mouse liver. a** UMAP representation based on gene expression of the spatial data. Cell type annotations obtained with NiCo are highlighted. The legend indicates the number of cells for each cell type. EC, endothelial cell; cDC, conventional dendritic cell; pDC, plasmacytoid dendritic cell; HSPC, haematopoietic stem and progenitor cell; ILC1, innate lymphoid cell type 1. **b** Select cell types are highlighted for a representative tissue area. Top, central vein and portal vein endothelial cells (EC). Bottom, hepatocytes of different zones, i.e., central (Hep C), mid-zonal (Hep M) and portal (Hep P1/P2). **c** Cell types from (**b**) are highlighted in the expression UMAP representation from (**a**). **d** Confusion matrix highlighting predictive capacity of the niche for all cell types. **e** Regression coefficients reflecting local niche interactions for stellate cells (left), central vein EC (center) and central hepatocytes (right). Regression coefficients (*y*-axis) were ordered by decreasing magnitude (*x*-axis). The red dotted line indicates the threshold for cell type interaction ($c = 0.15$) applied in (**f**). Error bars, see "Methods". **f** Cell type interaction map derived from the regression coefficients of NiCo's interaction module (Methods). Arrows point from predictive niche cell types towards the central cell type.

associate these covariates with full transcriptome variability. It is important to stress that cell state covariation inference requires mapping of individual mRNA molecules to the cell of origin. This is currently not possible with sequencing-based spatial transcriptomics, which require local pixel aggregation and subsequent deconvolution into cell types due to limited resolution and sensitivity. This process destroys local covariance structure between cell types in aggregated spots. With technological advancements, sequencing-based methods may be able to achieve single-cell resolution in the future, and NiCo will be directly applicable to this data type. Similarly, NiCo can be applied to future generations of full transcriptome imaging-based spatial technologies without requiring reference scRNA-seq data.

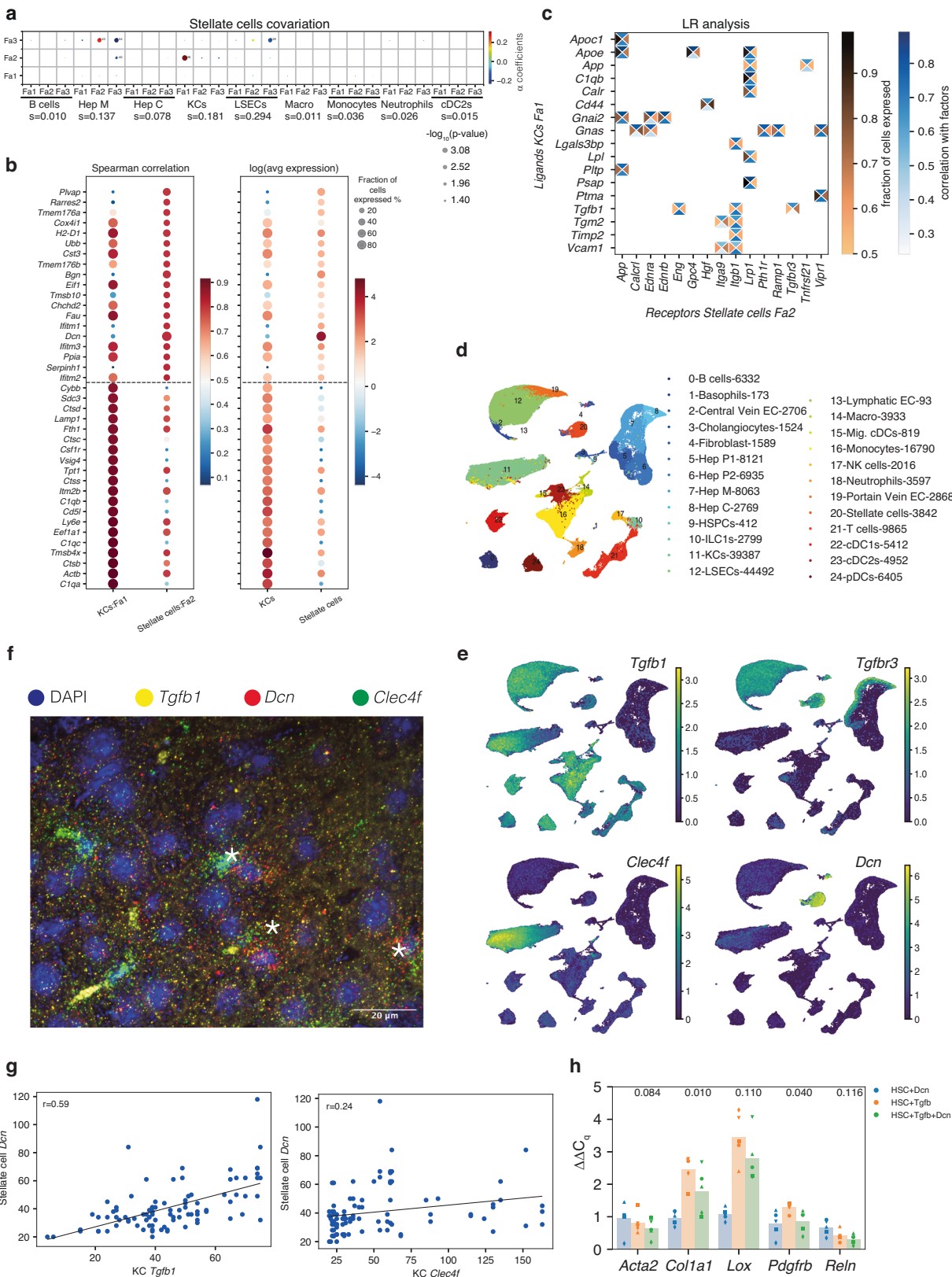

By applying NiCo to seqFISH data of E8.5 mouse embryos undergoing hierarchical dynamic organization into tissue domains, we showcase that NiCo identifies cell-cell interactions underlying the emergence of cardiac progenitors from the sub-pharyngeal mesoderm and recovers Wnt signaling as a critical pathway regulating this process. For mouse intestinal MERFISH data, NiCo provides deeper insights into the well-known Paneth cell niche of Lgr5+ intestinal stem cells, and associates Wnt activity with the Paneth cell progenitor state. In the mouse liver, NiCo captures cell state covariation between stellate cells and Kupffer cells, which could be essential for limiting stellate cell activation in the healthy liver, supported by Dcn-mediated dampening of HSC activation in vitro. These examples demonstrate that

**Fig. 6 | NiCo predicts crosstalk between stellate cells and Kupffer cells in the mouse liver. a** Covariation between stellate cell factors (*y*-axis) and co-localized neighborhood cell type factors (*x*-axis). Circle size scales linearly with -log$_{10}$(*p*-value), and circle color indicates ridge regression coefficients. S denotes the normalized niche coefficient score. The multivariate regression *p*-value was derived from two-tailed *t*-statistics. **b** Spearman correlations (left) and average expression (right) for the top 20 positively correlated genes from scRNA-seq reference data for Kupffer cell (KC) Fa1 and stellate cell Fa2. Dashed line demarcates the genes associated with each cell type. **c** Ligand-receptor pairs correlated with KC Fa1 and stellate cell Fa2. The rectangle's north and south faces represent ligand and receptor correlation to the factors, while west and east faces represent the proportion of ligand and receptor expressing cells. KC ligands, *y*-axis; stellate cells receptors, *x*-axis. See "Methods". **d**, **e** UMAP visualization of mouse liver cell types (d) and normalized expression of *Tgfb1, Tgfbr3, Dcn, Clec4f* (e). scRNA-seq/single-nucleus RNA-seq/CITE-seq data from Guilliams et al.[63]. Cell counts are indicated in the legend. **f** Single-molecule fluorescence in situ hybridization (smFISH) analysis of stellate cell and KC marker genes in mouse liver tissue sections (*n* = 3). Representative field of view with nuclei staining (DAPI, blue), and single mRNA molecule signals for *Tgfb1* (yellow), *Dcn* (red), and *Clec4f* (green). Asterisk, co-localized pairs of a stellate cell and a KC. **g** Scatterplot highlighting correlation of *Dcn* and *Tgfb1* (left), and *Dcn* and *Clec4f* (right) in co-localized stellate cells and KC. A regression line and Pearson's correlation coefficient (top left) are indicated. **h** qPCR quantification ($\Delta\Delta C_q$) of five genes (*n* = 5 biological replicates, datapoints correspond to averages across three technical replicates) for in vitro culture of HSC+Dcn, HSC +Tgfb, and HSC+Tgfb+Dcn conditions against control (HSC). Biological replicates are represented by different symbols. One-sided paired *t*-test *p*-values comparing HSC+Tgfb against HSC+Tgfb+Dcn for each gene are indicated on top of each bar. Source data of the qPCR and smFISH experiment are provided as Source Data file (see Data Availability).

NiCo can illuminate the effect of cell-cell interactions in local tissue environments.

NiCo relies on regularized regression analysis to identify predictive niche interactions and to infer factor covariation. Both steps are limited in their capacity to infer higher-order interactions and the covariation analysis is insensitive to strongly non-linear covariation patterns. While NiCo's regression framework in principle could accommodate for higher-order interaction terms, this would massively increase model complexity, which would be detrimental for robustness and interpretability. Future extensions should explore non-linear methods such as graph attention networks[78] for the exploration of higher-order interactions and non-linear niche covariation.

In summary, NiCo fills a critical gap for identifying spatial dependencies between cell states in common tissue niches and represents a significant step forward towards learning and interpreting extrinsic driving forces of cellular states. With the rapidly increasing availability of commercial platforms for high-resolution spatial transcriptomics we anticipate that NiCo will be a key component to gain biological insights from such data.

## Methods
### Dataset description
We analyzed spatial data and reference scRNA-seq of four different mouse tissues across seven studies.

**Organogenesis.** We analysed spatial transcriptomics data at single-cell resolution from mouse organogenesis generated by seqFISH[33]. The spatial data was captured during the embryonic time point E(8.5–8.75) and comprised 19,451 individual cells and 351 genes. For this dataset, 24 distinct cell types were annotated by the authors, which were used as ground truths for benchmarking cell type annotation. As reference dataset, a published scRNA-seq resource covering different stages of mouse organogenesis was used[41], which was subset to the E8.5 time-point corresponding to the spatial dataset. Cells designated as NaN cluster type were removed, resulting in 16,909 cells and 19,371 genes across 29 clusters. After merging the two blood progenitor clusters and three erythroid subtypes, 26 distinct cell types were identified by the authors, which were utilized for NiCo's annotation module.

**Small intestine.** We obtained single-cell resolution spatial transcriptomics data of the mouse small intestine generated with the MERFISH technology[31]. Corresponding scRNA-seq reference data of the same tissue were obtained from an independent study[54]. The published MERFISH annotation comprises 19 unique cell types, profiling 241 genes across 7416 individual cells. The scRNA-seq data encompass 32,287 genes, and 2239 cells that are stratified into two clustering partitions, i.e., high-resolution cell states and coarse cell types. Within the cell state category, 26 distinct cell states were identified, while the coarse category encompasses 17 unique cell types. To apply the NiCo annotation

module, we merged all populations of stem cell-related states, goblet-related cells, and stroma-related populations, respectively, resulting in 19 distinct cell type clusters. The intersection of the scRNAseq and MERFISH data comprises 240 common genes.

**Liver.** We utilized single-cell resolution spatial transcriptomics data of the mouse adult liver obtained with the Vizgen MERSCOPE platform (available from https://vizgen.com/data-release-program/). Corresponding scRNA-seq reference data from the Liver Cell Spatial Proteogenomics atlas were utilized[63]. The MERSCOPE dataset includes 347 genes and 391,679 individual cells, with cell filtration based on a minimum count of 5. The scRNA-seq atlas comprises 185,894 cells and 24,857 genes. We utilized the 'mouseStStAll' cell type annotation provided by the authors. This annotation was modified by merging specific immune cell types (CD8 Effector Memory T cells, CTLs, NKT cells, Naïve CD4 + T cells, Naïve CD8 + T cells, Th1s, TRegs, and Th17 cells into single T cell cluster; Patrolling Monocytes, Trans. Monocytes and Monocytes-derived cells into a single Monocyte cluster; and MoMac1, MoMac2, and Peritoneal Macrophages into a single Macrophage cluster) and by annotating subclusters of hepatocytes. The hepatocyte population in 'mouseStStAll' was divided into eight subclusters, which we categorized into four subpopulations of zonal hepatocytes based on their expression of zonation markers[61].

**Primary motor cortex.** We accessed MERFISH data of the mouse primary motor cortex[32] and corresponding scRNA-seq data[79]. The MERFISH dataset comprises 280,186 cells, encompassing 254 genes and 24 cell types as defined by the authors. These cells were sourced from 64 tissue slides derived from two mice litter. From this dataset, we selected a specific slide (mouse1, slice153) due to its continuous structure and the highest cell count, totaling 7626 cells. The scRNA-seq data, termed 'BICCN MOp scRNA 10X v3 Analysis AIBS' were obtained from the NeMo archive; we removed 'Low Quality' and 'doublet' subclass labels. This dataset consisted of 71,183 cells and 30,982 genes, and we retained 20 subclass labels as defined by the authors. Nineteen of these cell types mirrored those identified in the MERFISH data.

**Cerebellum Slide-seqV2.** The Slide-seqV2 mouse cerebellum dataset[73] comprises 44,091 cells and 5160 genes, with cell types annotated using the RCTD[80] method by the authors. We retained the singlet and doublet_uncertain categories, resulting in 30,575 cells and 18 cell types. The NiCo interaction module was applied to this dataset with default parameters and the covariation module was applied with the unimodal version of ordinary NMF.

**STARmap mouse visual cortex dataset.** STARmap mouse visual cortex data[18] comprise 1207 cells and 1020 genes, 16 cell types, annotated into 7 neocortical layers L1, L2/3, L4, L5, L6, HPC (hippocampus) and cc (corpus callosum).

**MERFISH mouse brain atlas**. The MERFISH atlas of the whole mouse brain[40] comprises 59 brain sections with ~4 million cells. We analyzed z-slice 11.2, containing 44,699 cells and 500 non-blank genes, falling into 17 cell type classes and 6 neighborhood classes with regional specificity. The six neighborhood classes are denoted as D1 (HY-EA-Glut-GABA), D2 (NN-IMN-GC), D3 (Pallium-Glut), D4 (Subpallium-GABA), D5 (Subpallium-GABA; HY-EA-Glut-GABA), D6 (Subpallium-GABA; NN-IMN-GC).

## Niche Covariation (NiCo) algorithm

**The NiCo cell type annotation module.** We adopted a multistep approach to integrate single-cell resolution spatial transcriptomics with scRNA-seq reference data, assuming a given cell type annotation of the reference data available as clustering partition. We first give a brief overview of the pipeline, followed by a detailed explanation of each step: (1) *Determining anchors*: Following preprocessing, NiCo first identifies anchor cells between the spatial query data and the scRNA-seq reference data by the mutual nearest neighbor (MNN) method[81]. (2) *Anchor filtering*: We made use of gene expression-based clustering of the spatial data using the Leiden community detection algorithm to filter out dispersed anchors mapping to reference clusters at low frequencies. (3) *Iterative annotation of non-anchors*: Non-anchor cells in the spatial query data receive a cell type label based on the cell type frequency of anchors among their K-nearest neighbors (KNN) and join the anchor set. This process is iterated.

**Determining anchors.** NiCo's annotation module takes cell-by-gene count matrices of query and reference data, denoted as $M_q$ and $M_r$, as input data. After sub-setting to the shared set of genes measured in both modalities, normalization using the scTransform method[82] yields Pearson's residuals for downstream analysis. Next, we extracted the top 50 principal components (PCs) from each normalized matrix, transforming them into latent dimension matrices, $d_q$ and $d_r$. These 50 PCs retained the majority of the variance in the data. We standardized $d_q$ and $d_r$ and employed a KD-tree algorithm to find mutual nearest neighbor (MNN) pairs applying a soft criterion. Specifically, we identify for each query cell $q$ the set of KNN in the reference data, termed $s_q$. Similarly, we identify for each reference cell $r$ the set of KNN in the query data, termed $s_r$. By default, we use K = 50. Query cell $q$ and reference cell $r$ are considered MNN, if $q \in s_r$ and $r \in s_q$. This neighborhood-based definition of MNNs permits for the existence of multiple mutual nearest neighbor cells in the scRNA-seq reference dataset for a given query cell. In this case, the query cell is considered a "confused anchor". This is resolved by drawing information from the non-confused anchor cells among the KNNs (K = 50 by default) of the query cell. After pruning neighboring anchors that do not belong to the same spatial guiding cluster (see "Anchor filtering") as the query cell, a unique cell type is assigned based on the largest proportion of pruned neighboring cells by majority vote. If this results in a tie or if no filtered neighboring cells belong to the same spatial guiding cluster as the query cell, it is designated 'NM' (Not mapped). If this procedure results in too many cells that could not be mapped, we implemented a weighted score instead of a majority vote for cell type assignment based on neighboring anchor information. The weighted score is computed based on the inverse distance between the confused query cell and its neighboring anchors after pruning. Scores are aggregated across neighbors with the same assigned cell types, and the cell type with the largest score is assigned to the query cell.

**Anchor filtering.** For each pair of MNN the cell type label is transferred from the scRNA-seq reference annotation to the spatial query data. We next perform low resolution Leiden clustering of the spatial query data $d_q$ to infer clusters for anchor pruning (spatial guide clustering). These guiding clusters are utilized to filter out noisy anchors. Anchors of a given cell type label are considered noisy, or dispersed, if they map to a

spatial cluster with a frequency lower than the dispersion parameter $r$ ($r = 0.15$ by default). This step implements information sharing between anchors of the same cell type label by keeping anchors falling into the same spatial guiding cluster at high frequency and removing anchors falling into spatial guiding clusters at low frequency. The resolution of the spatial guide clustering can be adjusted; lower values retain more anchors but may decrease specificity of the annotation.

**Iterative annotation of non-anchors.** Anchor cells serve as a seed for annotating the remaining non-anchor cells in the spatial query data. Each non-anchor cell is annotated by a majority vote based on cell type proportions of anchor cells among its KNN. If the cell type with the highest proportion by majority vote and the queried non-anchor cell end up in same guiding cluster, the queried non-anchor cell is assigned to this cell type. Moreover, if a query cell's majority vote for cell type assignment results in a tie, NiCo marks it as 'NM' (Not Mapped). This process is iterated, and after each iteration anchors are updated with the annotated non-anchors. This process is only iterated three times to avoid low confidence annotations. NiCo successfully mapped ~96% of query cells in embryo and liver data, ~99% of query cells in intestine data, and ~90% of query cells in brain data. To further reduce the number of non-mapped cells, cell type annotations of neighbors can optionally be weighted by their relative distances as explained in the step 1 to the query cell, which avoids ties but may reduce specificity and is therefore not used as default setting.

**Benchmarking cell type annotations.** In our comparative analysis, we benchmarked NiCo against four state-of-the-art methods: cell2location[27], Tangram[28], TACCO[29] and uniPort[30]. The evaluation was conducted based on two primary criteria: the adjusted rand score (ARI) and Jaccard similarity (JAC). The author's annotation of each dataset was considered as ground truth. To compute these metrics, the cell type names for the reference annotation had to match exactly the author-defined annotation. We identified 14, 20, and 17 matched cell types for intestine, embryo, and cortex data, respectively, out of 19, 26, and 20 total scRNA-seq reference clusters. Initially, all datasets were preprocessed with a filtering step, setting the minimum total transcript count to 5 in order to eliminate low quality cells. For NiCo, the resolution parameter for spatial guide Leiden clustering was set to 0.4 for intestine, cortex, and embryo datasets, and 0.5 for the liver data.

Tangram was executed with default parameters, mapping cells to space using the 'clusters mode' and reference cluster labels from scRNA-seq data. uniPort was run with default parameters, employing label transfer from reference data to query data for spatial cell type annotations. TACCO was run with default parameters. For cell2location, parameters "N cells per Location=1" and "detection alpha=20" were applied, with all other parameters set to their default values. To avoid memory overflow, cell abundance was estimated from the posterior distribution summary in the large liver dataset by directly using quantile computation. For other datasets, the quantile computation uses 1000 "num_samples" to generate a posterior distribution and uses the 0.05 quantiles to obtain spatial cell type annotations, following the developer's recommendation.

**Calculation of computation time and memory usage for cell type annotation and niche prediction tasks.** The computation time and memory requirement for different tasks were measured using Python's time and memory_profiler modules. Specifically, these module were called at the beginning and end of the script to calculate the total execution time and memory usage.

**The NiCo interaction module.** To infer significant cell type interactions based on spatial co-localization, NiCo employs a logistic regression classifier to predict the cell-type identity of a "central" cell from the cell type composition of its neighborhood. In short, the method

proceeds in three steps. (1) *Inference of niche composition*: We first determine the local-niche neighbors for each cell. Subsequently, we infer global expected cell type frequencies and determine fold enrichments for every cell type in each neighborhood. (2) *Logistic regression classifier:* We then train a logistic regression classifier to predict the central cell type identity from the cell type enrichment ratios in the local niche. (3) *Cell type interaction graph:* The regression coefficients are then utilized to construct a cell type interaction graph.

**Inference of niche composition.** Local neighborhood information is deduced based on cell centroids. Two distinct schemes are employed for defining neighborhood composition. The first scheme is based on juxtacrine signaling and includes only direct neighbors. This approach, also referred to as 'radius=0', is executed through Delaunay triangulation. Since Delaunay triangulation can include distant neighbors due to wide intercellular space in the tissue, we define a cutoff of 100μm and discard neighbor with a centroid-to-centroid distance beyond this limit. The second scheme accommodates paracrine signaling by including all neighbors whose centroid is localized within a given radius $R$ around the central cell's centroid.

Once the neighbors are identified, we create a cell type count matrix $\Lambda_{ij}$ and a vector $y_i$ with $i = 1,...,m$ and $j = 1,...,n$. $n$ corresponds to the number of cell types, and $m$ is equal to the total number of cells in the dataset. $\Lambda_{ij}$ denotes the count of cell type $j$ in the neighborhood of the central cell $i$. The cell type identity of cell $i$ is stored in $y_i \in (1,..., K)$. The number of classes $K$ corresponds to the number of cell types and is therefore identical to the number of features $n$. The counts $\Lambda_{ij}$ are normalized by expected global cell type frequencies to obtain the neighborhood enrichment matrix

$$X_{ij} = \Lambda_{ij} / \sum_k f_j \cdot \Lambda_{ik} \tag{1}$$

where $f_j$ corresponds to the global frequency of cell type $j$ and $\sum_j f_j = 1$.

**Logistic regression classifier.** NiCo employs a multi-class logistic regression classifier to learn a set of coefficients reflecting the relevance of each niche cell type for predicting a given central cell type. This classifier computes the probability $P(y_i = k, |, \mathbf{x}_i; W)$ of central cell $i$ to be of cell type $k$ given the enrichment ratios $\mathbf{x}_i$ of all cell types in the neighborhood of cell $i$:

$$P(y_i = k | \mathbf{x}_i; W) = \frac{\exp(\mathbf{w}_k \mathbf{x}_i + w_{0,k})}{\sum_{j=1}^{K} \exp(\mathbf{w}_j \mathbf{x}_i + w_{0,j})} \tag{2}$$

$\mathbf{x}_i$ corresponds to the $i$-th row of the neighborhood enrichment matrix $X$ and $\mathbf{w}_k$ corresponds to the $k$-th column of $W$.

The algorithm incorporates an L2 penalty for regularization, with its hyperparameter, denoted as $C$, determined using the GridSearchCV function of the scikit-learn python library by minimizing the negative log-likelihood

$$\min_w \left( -C \sum_{i=1}^{n} \sum_{k=1}^{K} [y_i = k] \log(P(y_i = k | \mathbf{x}_i; W)) + \frac{1}{2} \sum_{i=1}^{n} \sum_{j=1}^{K} w_{j,i}^2 \right) \tag{3}$$

Stratified 5-fold cross-validation is applied. The input feature matrix $X$ is normalized using the StandardScaler function. Furthermore, class weights are adjusted using the balanced mode to account for class imbalance. The 'class_weight' function of the scikit-learn python library assigns higher weights to minority classes, allowing the model to pay more attention to its patterns and reducing bias towards majority classes. In this mode, values of $y$ adjust weights inversely proportional to class frequencies in the input data.

The L-BFGS method implemented in the 'lbfgs' solver function of the scikit-learn python library is employed during the training step,

alongside the 'multinomial' logistic loss function encompassing the entire probability distribution.

The vector $\mathbf{w}_i^f = \left(w_{i1}^f, ..., w_{in}^f\right)$ corresponds to the $i$-th column of the coefficient matrix $W^f$ for cross fold $f$ ($f = 5$), i.e., contains the cell type coefficients for predicting each central cell type $i$ from its niche composition $x_i$. It is averaged across all data cross folds $f = 1, ..., F$:

$$\boldsymbol{\beta}_i = \frac{1}{F} \sum_{f=1}^{F} , \mathbf{w}_i^f \tag{4}$$

The standard deviation of $\mathbf{w}_i^f$ reflects the uncertainty of the coefficients and serves to estimate error bars. For follow-up analysis we recommend disregarding coefficients with error bars of similar magnitude to the coefficient value itself.

To assess the predictive capacity of the niche composition for each cell type, we inspect the confusion matrix, which compares the predicted cell type label to the ground truths label corresponding to the NiCo annotation of the central cell. We report the average confusion scores across data folds and normalize each row of the confusion matrix (ground truths cell type labels) to one.

To facilitate the visualization of the coefficients for each central cell type $i$, we plot the mean $\boldsymbol{\beta}_i$ in descending order of their absolute value along the $x$-axis and indicate uncertainty by the standard deviation across data folds. Positive coefficients $\beta_{ij} > 0$ indicate that the neighboring cell type $j$ preferentially co-localizes with central cell type $i$, whereas negative coefficients $\beta_{ij} < 0$ signify preferential absence of cell type $j$ from the niche of cell type $i$.

**Cell type interaction graph.** To create a cell type interaction map highlighting cell type co-localization patterns, we normalize $\boldsymbol{\beta}_i$ by dividing it by the maximum absolute value of its components $\beta_{ij}$. After applying a cutoff on $\beta_{ij}$, we draw edges connecting the central cell type $i$ and niche cell type $j$ with directionality pointing from $j$ to $i$, resulting in a graph structure comprising all cell types and their interaction partners. A global cutoff can be applied to $\beta_{ij}$ to control the sparsity of the graph. The edge thickness scales with $\beta_{ij}$. We employed a force-directed layout within the framework of the Graphviz layout.

**Testing the NiCo interaction module on simulated ground truths data.** To test the capacity of NiCo's logistic regression classifier to predict the central cell type's class from the cell type neighborhood, we conducted a simulation, where we created six cell types, each containing 200 cells, in a 2D space using the LAMMPS molecular dynamics package[83]. This simulation was carried out with a varied Lennard-Jones (LJ) pairwise potential for modeling interactions between cell types. LJ potentials describe interactions that are repulsive on short range (particle size σ), to simulate physical cell boundaries, and attractive at longer distances with a given range (interaction cutoff range parameter rc) and strength (potential well ε), to simulate preferential co-localization of defined pairs of cell types.

Fig. 2c: We first initialized all (6 + 6 choose 2) = 21 pairwise interactions, including self-interactions, with the following parameters: potential well ε = 1, particle size σ = 2, and interaction cutoff range parameter rc = 5. Subsequently, we introduced stronger interaction potentials between T0-T2 (ε = 3) and T3-T5 (ε = 5). This allowed us to create a scenario where T0 was expected to be in the neighborhood of T2, and vice versa, and T3 should be in proximity to T5, and vice versa. The simulation box had dimensions of −50 to +50 in both X and Y, and periodic boundary conditions were assumed. All masses were set to 1, initial velocities were assigned to achieve a temperature of T = 1 at the normalized kB unit, and equilibration was attained using NVE and Langevin's thermostat. This ensured that each cell maintained the desired temperature of T = 1, and the enforce2d command was employed to constrain cell movement along the Z direction to mimic

the simulated locations in the 2D plane. The final configuration of the simulation served as the input for NiCo to predict neighborhood niche interactions. The confusion matrix and coefficients for predicting each cell type were computed for the default parameter of radius=0, considering only immediate neighbors.

Fig. 2d: To test a more challenging situation, we simulated a more complex scenario. All six pairwise interactions, including self-interactions, were initialized with parameters ε = 1, σ = 2, and rc = 5. We then altered the ε parameters for T0-T2, T3-T5, T2-T3, and T1-T3 interactions to 3, 5, 10, and 8, respectively, to establish a distinct order among the cell types. The confusion matrix and coefficients for predicting each cell type were computed for the default parameter of radius=0, considering only immediate neighbors.

**Benchmarking the NiCo interaction module with MISTy.** In our comparative study between NiCo and MISTy[22] for niche prediction, we transformed cell types for the more complex, second simulated scenario into a one-hot encoding scheme. Subsequently, we input these data into MISTy, generating a juxtaview with default parameters and paraview with radius five. Inferred importance, i.e., z-transformed total variance reduction scaled by (1 - *p*-value), was interpreted as indicating cell type interactions between a central cell type (target) and niche-cell types (predictors).

**Detecting covariation of gene expression in co-localized cells with NiCo.** To detect covariation of co-localized cell states in local niches, NiCo's covariation module implements the following steps: (1) *Latent variable inference*: Identification of a set of latent variables for each cell type that capture intra-cell type variability. (2) *Latent variable annotation*: Inference of genes and pathways correlated to the latent factors, including ligand and receptor pairs to predict signaling mediators. In this step, we focus on the scRNA-seq data to obtain full transcriptome coverage. (3) *Identification of niche covariation*: NiCo applies regularized linear regression to identify latent factors in co-localized cell types exhibiting covariation.

**Latent variable inference.** NiCo employs an integrated Non-Negative Matrix Factorization (iNMF) approach alongside ordinary NMF (oNMF) to identify latent factors from the shared set of genes in the spatial transcriptomics and the scRNA-seq data. Both datasets are scaled but not mean-centered. These NMF methods learn a lower-dimensional embedding space, encompassing spatial cell factors $H_{sp} \in \mathbb{R}_+^{c_{sp} \times LF}$, scRNA-seq cell factors $H_{sc} \in \mathbb{R}_+^{c_{sc} \times LF}$, and shared gene embedding factors $W \in \mathbb{R}_+^{LF \times G}$, where *LF* corresponds to the user-defined number of factors, *G* denotes the number of shared genes across modalities, and $c_{sp}$ and $c_{sc}$ denote the number of cells for a particular cell type in the spatial and the scRNA-seq data, respectively. The iNMF methodology[25,26] draws inspiration from the LIGER package[25,26], and within NiCo, it simultaneously learns common gene factors $W$, $H_{sc}$ factors and $H_{sp}$ factors from cell type-specific scRNA-seq reference cluster $E_{sc}$ and query spatial cluster $E_{sp}$ after sub-setting to the shared set of genes. The objective function of iNMF is designed to learn a shared factor $W$ across all datasets while capturing data heterogeneity through factors $H_x$ and $V_x$ with *x* denoting the modality:

$$\text{argmin}(||E_{sc} - H_{sc}(W + V_{sc})||_F^2 + \lambda|| H_{sc}V_{sc} ||_F^2$$

$$+ ||E_{sp} - H_{sp}(W + V_{sp})||_F^2 + \lambda||H_{sp}V_{sp}||_F^2) \quad (5)$$

Optimization incorporates penalizing the Frobenius norm (denoted by *F*) after accounting for heterogeneous effects across modalities, $H_{sc}V_{sc}$ and $H_{sp}V_{sp}$, and regularization of these components with parameter $\lambda$. The matrix product $H_xV_x$ encodes data-specific constraints for modality *x* tailored for heterogenous multi-modal data,

following the principles outlined in ref. 26. Importantly, all learned matrices, $H_x$, $V_x$, and $W$, are constrained to be non-negative. The selection of $\lambda$, a tuning parameter, is critical and preferably small for a mixture of homogenous datasets and larger for a mixture of heterogeneous datasets. To determine the optimal $\lambda$ from $E_{sc}$ and $E_{sp}$, we execute iNMF for increasing values of $\lambda$ set to even integers ($\lambda = 0, 2, 4, 6, \ldots$), and selecting the $\lambda$ that stabilizes the alignment score (Eq. (6)), following the methodology outlined in ref. 84. In brief, we first randomly downsampled the larger dataset ($H_{sc}$ or $H_{sp}$) to match the size of the smaller dataset and merged them into a unified dataset. We determine $K$ as 1% of the total number of cells in the smaller dataset, denoted as $n$, and identified the $K$-nearest neighbors for each cell in the unified dataset. For every cell, we calculated how many of its KNN belonged to the same dataset and average this across all cells to obtain $\bar{x}$. In an ideal scenario where $H_{sc}$ and $H_{sp}$ are learned perfectly, each cell's K-nearest neighbors would be evenly distributed across the two datasets. We then normalized this value by the expected number of cells from the same dataset and scaled it to range from 0 to 1. Ultimately, the alignment score between two successive $\lambda$ iterations was considered stable when the difference between them fell below the threshold of 0.001, determining our final $\lambda$.

$$\text{score} = 1 - \frac{\bar{x} - \frac{K}{n}}{K - \frac{K}{n}} \quad (6)$$

For the MERSCOPE liver dataset, we noticed intermingling of all cell types in the dimensional reduction representation of the spatial data (Fig. 5a). Potential explanations for this observation are imperfect cell segmentation or high background. This leads to a challenge during the factorization process of iNMF, since factors could become strongly associated with spill-over and/or background genes, which would not be indicative of their actual prevalence within certain cell types. To address this issue, we implemented an alternative approach known as ordinary NMF (oNMF). In oNMF, the matrices $H_{sc}$ and $W$ are only learned from reference data $E_{sc}$. We use the iterative element-wise multiplicative update solver[85]. The objective function in oNMF aims to minimize the 'Kullback-Leibler' divergence, denoted as $d_{KL}$, which measures the dissimilarity between matrices $X$ and $Y$: $d_{KL}(X,Y) = \sum_{i,j}(X_{i,j}\log(\frac{X_{i,j}}{Y_{i,j}}) - X_{i,j} + Y_{i,j})$. We initialize the matrices $W$ and $H_{sc}$ using the nndsvda method of the scikit-learn python library (Non-negative Double Singular Value Decomposition with zeros filled with the average of $E_{sc}$). Our optimization process runs for a maximum of 1000 iterations, and no regularization parameters are applied to matrices $H_{sc}$ and $W$. Following the derivation of $W$ from the scRNA-seq data, we fix these factors for the spatial data while optimizing $H_{sp}$. This is achieved through an analogous element-wise iterative multiplicative update rule keeping $W$ constant:

$$H_{sp}[i,j] = \frac{H_{sp}[i,j] * (W^T E_{sp}^T)[i,j]}{(W^T W H_{sp})[i,j]} \quad (7)$$

Importantly, we do not update the matrix $W$ during this process. This strategy enables us to infer spatial cell factors reflecting the gene expression variability information contained in the scRNA-seq data.

The number of factors $LF$ per cell type is a user-defined parameter. Since cell type populations are already homogenous and intra-cell type variability is typically limited, we execute NiCo with a small number of factors per cell types ($LF = 3$ by default). By performing consensus-NMF analysis[86] on the intestinal datasets, we found that factor stability frequently drops significantly for larger numbers of factors, and tuning the number of factors individually for each cell type would require substantial user intervention. We recommend running NiCo with three to five factors per cell type.

The factors obtained from NMF lack an inherent order. To address this, we introduce a quantification of disorder within each factor $i$ of cell type $k$, referred to as entropy:

$$E^k[i] = \frac{-\sum_i(H_{sc}^k[:,i]*\log_2(H_{sc}^k[:,i]))}{\log_2\left(\text{length}(H_{sc}^k[:,i])\right)} \quad (8)$$

This normalized entropy scales between 0 and 1. We calculate the entropy of scRNA-seq cell factors ($H_{sc}$) and reorder the factors by increasing entropy. Subsequently, the same ordering is applied to the spatial cell factors ($H_{sp}$). We found that the normalized entropy of iNMF-derived factors is typically higher than for oNMF-derived factors. This sorting process provides a more structured and informative representation of the factors.

**Latent variable annotation.** Finally, both NMF approaches yield a factors-by-genes matrix $W$; each factor thus corresponds to a linear combination of genes, representing specific gene modules or meta-genes for each cell type. Since factors are derived from the shared set of genes, they do not provide transcriptome-wide information. To conduct comprehensive gene covariation analysis, pathway enrichment, and ligand-receptor analysis, we require information associating the whole transcriptome with each of these factors. To achieve this, we utilize two distinct metrics, the Spearman correlation and cosine similarity. Specifically, the Spearman correlation measures the association of each gene with every factor by calculating correlations between each column of $E_{sc}$ (representing cell type-specific scRNA-seq whole gene expression profiles) to each column of $H_{sc}$. Accordingly, the cosine similarity is obtained by taking the dot product between a column of $E_{sc}$ and the column of $H_{sc}$. Top positively or negatively correlating genes for each factor could then be inspected and functionally annotated, e.g., by pathway enrichment analysis. Ligands and receptors among these genes can be nominated for inferring inter-cellular signaling underlying observed covariations of factors between cells.

**Identification of niche covariation.** We applied regularized linear regression analysis to elucidate covariation between latent factors of co-localized cell types residing in the same tissue niche. NiCo explores these relationships between factors corresponding to a central cell type and those pertaining to co-localized neighboring cell types as identified by the interaction module. By design, NiCo is sensitive to approximately linear relationships and returns regression coefficients between the dependent variable (representing factors of central cell types) and the independent variables (comprising weighted averages of factors from neighboring cell types).

Given a central cell type $k$, we include all neighboring cell types $j$ occurring across instances of cell type $k$ with regression coefficients $\beta_{kj} > c$ for a given cutoff $c$. By increasing $c$, neighboring cell types of limited predictive capacity as inferred by NiCo's interaction module can be discarded.

For this analysis, let $H_{sp}^k[j,i]$ denote the $i$-th factor of the $j$-th instance of the central cell type, and consider cell factors $H_{sp}^m$ of neighboring cell types $m \in \{1, \ldots, M\}$ with $\beta_{km} > c$. We infer covariation between $H_{sp}^k[,i]$ and all factors $H_{sp}^m$ across all instances $j$ by ridge regression:

$$\min_\alpha \left( \left\| \sum_m \sum_{l=1}^{LF} \alpha_{l,m}^{k,i} \frac{1}{N_j^{k,m}} \sum_{h \in S_j^{k,m}} H_{sp}^m[h,l] - H_{sp}^k[j,i] \right\|_2^2 + \eta \left\| \alpha_{l,m}^{k,i} \right\|_2^2 \right) \quad (9)$$

Here, $\alpha_{l,m}^{k,i}$ denotes the regression coefficients between factor $i$ of the central cell type $k$ and the latent factor $l$ of neighboring cell type $m$.

$S_j^{k,m}$ denotes the set of all cells of cell type $m$ in the neighborhood of the $j$-th instance of central cell type $k$ and $N_j^{k,m}$ denotes the number of these cells. Hence, NiCo averages a latent factor across all cells of a given neighboring cell type.

The ridge regression regularization parameter $\eta$ is optimized through the following procedure: The initial $\eta$ parameter range spans powers of 2, ranging from $[2^{-10}, 2^{-9}, \ldots, 2]^{10}$. GridSearchCV calculates the negative mean squared error using the leave-one-out cross-validation strategy to determine the optimal $\eta$ parameter.

## Ligand-receptor interaction analysis

To identify ligand-receptor signaling interactions as potential mediators of the inferred covariation of factors in co-localized cells, we first compiled a ligand-receptor database from available resources. We created NiCoLRdb by merging ligand-receptor pairs from Omnipath[87], NATMI[88], and CellPhoneDB[89,90]. Multimeric complexes from CellPhoneDB were not included. From this database we extract ligand-receptor (LR) cognate pairs within the cell type of interest and inter-acting neighboring cell types included in the covariation analysis. Since only limited numbers of ligand and receptor genes are typically directly measured in the spatial data, we leverage information from the genome-wide scRNA-seq data. For a pair of interacting cell types with covarying factors inferred by NiCo, we only retain ligand-receptor pairs where the ligand and receptor genes are expressed in a minimum fraction $f_{LR}$ of cells for the interacting cell types. Furthermore, we require a minimum absolute Spearman correlation $c_{LR}$ of the ligand and receptor genes with the respective factor in the covarying cell type. To increase sensitivity, $f_{LR}$ and $c_{LR}$ can be lowered, and higher values of these thresholds select candidates with stronger associations with the covarying factors. However, it is not necessarily expected that both ligand and receptor correlate with the covarying factors. For example, the receptor could be constitutively expressed in the receiving cell type, while the ligand is induced together with a gene program associated with a factor in the sending cell type. Therefore, it is recommended to apply low cutoff values for $f_{LR}$ and $c_{LR}$ and visually inspect the metrics for each pair. For visualization, we devised a quadrilateral (rectangle) plot uniting the fraction of expressing cells and factor correlation values within a dot-plot-like visualization to highlight LR crosstalk. Four isosceles triangles form this unit, each oriented towards one of the rectangle's edges and joined in the center. The western and eastern faces represent the fraction of cells expressing the ligand and receptor, respectively, while the northern and southern faces highlight ligand and receptor factor correlation values. Ligands are consistently mapped on the y-axis, while receptors are denoted on the x-axis.

## Pathway enrichment analysis for factor-associated genes

We conducted pathway enrichment analysis using the Enrichr module within the GSEApy package[91]. The objective was to assess whether a predefined set of genes displayed statistical significance or relevance with respect to specific gene sets or pathways from the Enrichr library. We supplied the top positively or negatively correlated genes for a given factor as our input 'gene list'. This gene list was then queried against various Enrichr libraries, such as 'BioPlanet 2019', 'Reactome 2016', or 'GO Biological Process 2021' to assess the enrichment within diverse cellular processes or pathways. Enrichr visually presents significant cellular processes as a dot plot, where the $Y$-axis represents these processes as a function of a combined score. The size of each dot indicates the percentage of genes shared within specific cellular processes, while the color code reflects the associated $p$-value. The combined score, denoted as '$c$' is defined as the product of the logarithm of the $p$-value and the z-score. The $p$-value is calculated using the Enrichr Fisher's exact test, while the z-score is determined through the Enrichr correction applied to the Fisher's exact test[92].

## smFISH experiment

**Sample preparation and image acquisition.** Single molecule FISH by smHCR was performed on 5 μm cut, snap-frozen, methanol fixed mouse liver tissue sections from C57B/6 J males with an age of 12–16 weeks ($n = 3$). Coding-sequences of *Clec4f* (NM 016751.3), *Tgfb1* (NM 011577.2), *Dcn* (NM 001190451.2) were used to design smFISH probes of 20 nt length. Probes were designed by Molecular Instruments (Los Angeles, United States) and the hybridization protocol was performed according to the manufacturer. Samples were imaged using an Olympus IX83 CSUW1-TS2 spinning disc confocal microscope equipped with 405, 488, 561 and 640 nm LED lasers using a 60x oil immersion objective. Image acquisition was performed with a resolution of (149.76 × 149.76) microns equivalent to 2304 × 2304 px (binning 1 × 1) and images opened with the Bio-Formats Importer and further processed using FIJI[93].

**Detection of RNA spots and cell segmentation.** We employed the STARFISH package[94] for finding the precise localization of RNA spots and harnessed the QuPath software[95] to perform the cell segmentation based on extended DAPI regions with default parameters. Our smFISH data acquisition leveraged four lasers, each capturing distinct molecular signals across different areas of tissue into four channels: Tgfb1(Red 561 nm laser), Dcn (Far-red 639 nm laser), Clec4f (Green 488 nm laser), and DAPI (Blue 405 nm laser). We initiated our analysis with a white top hat filter to remove the background noise with a masking radius of 3. Subsequently, we normalized and enhanced each channel's intensity using the STARFISH 'ClipPercentileToZero' function within the percentile intensities parameter range of p min = 80 to p max = 100. To ensure the robustness of detected spots, we wrote a custom script to convert detected spot coordinates into Tif images and then systematically overlaid the Tif images of raw signals. This way we verified detected spots for a range of parameters in FIJI[93] for each channel across different fields of view (FOV). Once we achieved a satisfactory level of spot detection, we proceeded to select optimal parameter values. These included the minimum and maximum sigma parameters (1.5, 3), numerical sigma (10), and the intensity threshold parameter (95th percentile), facilitating spot detection across all FOVs. We saved all the detected spot centroid parameters (Xc, Yc) and the radius parameter r of the spots for further analysis.

**Analysis of expression correlation in co-localized pairs of stellate cells and Kupffer cells.** The cell segmentation data, derived from QuPath, were processed using the regionprops command from the scikit-image python library. This command stores the pixel coordinates of the true cell morphological region as a labeled mask. We generated discrete region masks associated with individual spots bounded in x dimension within the interval [ceil(Xc − r), floor(Xc + r)] and bounded in y dimension within [ceil(Yc − r), floor(Yc + r)]. By assessing the containment of a spot's mask within a cell's mask, we effectively determined the complete overlap of the spots region with the cell region. Spots failing to satisfy this criterion were excluded from further analysis. This process was iteratively applied, resulting in a tally of associated spots for each cell across the Clec4f, Dcn, and Tgfb1 gene transcripts.

To ascertain whether the cell's identity belongs to the Kupffer cell (KC) or stellate cell population, we first selected all the cells exhibiting over 20 transcripts for either *Dcn* or *Clec4f*, along with satisfactory segmentation accuracy. The maximum number of expressed transcripts in a single cell could be over 150 in such a setting. We labeled cells as stellate cells if *Dcn* transcript counts exceed *Clec4f* transcript counts, and as KC if *Clec4f* transcript counts are greater than *Dcn* transcript counts. This soft criterion accounts for imperfect segmentation due to the difficult-to-segment shape of stellate cells with long protrusions. Cells not meeting these criteria remained unclassified. To explore spatial relationships, we constructed cell neighborhoods

based on Delaunay Triangulation using the centroid positions. Complex units comprising adjacent KC and stellate cells were identified from the colocalized neighborhood. In total, 93 such complex units were discerned for three FOVs. From all complex units, we generated a scatter plot, with *Dcn* transcript levels from stellate cells represented on the *y*-axis and *Tgfb1* transcripts from KC on the *x*-axis. Pearson correlation coefficients were computed, and linear regression was applied for quantification.

## Hepatic stellate cell culture experiment

**Animal work.** Male C57BL/6 mice were used and all the animal experiments that were performed, were in accordance with the local and institutional regulations for the Protection of Animal Welfare (Regierung Unterfranken, Bavaria, Germany).

**Isolation and cultivation of mouse hepatic stellate cells (HSCs) and real-time PCR.** HSCs were isolated from 25 to 35 weeks old, male C57BL/6 mice as previously described[96]. In brief, in situ perfusion with EGTA buffer and Collagenase solution was followed by a successive Collagenase and Pronase in vitro digest. HSCs were washed and enriched by non-ionic gradient centrifugation and seeded in 24 wells (6 × 10⁴ cells per well; Thermo Scientific, Waltham, United States). Prior to cell stimulation, HSCs were cultured for 72 h in DMEM (Thermo Scientific) supplemented with 10% FBS (Corning, New York, United States) and 1% penicillin–streptomycin (Thermo Scientific). To activate Tgf-β signaling, cells were washed with PBS (Thermo Scientific) and treated with recombinant mouse TGF-β1 (10 ng/ml; Biolegend, San Diego, United States), under FBS free conditions and compared with pretreated HSCs, which received recombinant mouse Decorin (10 μg/ml; R&D Systems, Minneapolis, United States) for 1 h before TGF-β1 was added. After 72 h of treatment, cells were harvested for RNA extraction, cDNA synthesis and qPCR analysis, as previously described[97]. Total RNA was extracted using Arcturus PicoPure RNA Isolation Kit and cDNA was synthesized using RevertAid First Strand cDNA Synthesis kit (both Thermo Scientific) according to the manufacturer's instructions. For qPCR data analysis, the relative gene expression values for the target mRNA were normalized to *Gapdh*, which was selected from a set of 3 housekeeping genes (*Gapdh, Hprt, Ppia*), based on the normqPCR algorithm[98]. Used primers are listed in Supplementary Table 1.

**Analysis of qPCR data.** We performed three technical replicates and five biological replicates for the experiment, measuring the $C_q$ (Quantification Cycle) values of *Acta2, Col1a1, Lox, Pdgfrb*, and *Reln* genes. For each biological replicate, we first averaged the technical replicates and quantified the relative expression of each gene $i$ of interest using: $\Delta c_{qi}^j = 2^{(C_{q\text{Gapdh}}^j - C_{qi}^j)}$, where $j$ represents the condition from the following groups: control (HSC), HSC+Tgfb, HSC+Dcn, and HSC+Tgfb+Dcn. We then quantified the relative gene expression over the control group: $\Delta\Delta c_{qi}^j = \frac{\Delta c_{qi}^j}{\Delta c_{qi}^{\text{control}}}$. Finally, we performed a two sample paired t-test between $\Delta\Delta c_{qi}^{\text{HSC + Tgfb}}$ and $\Delta\Delta c_{qi}^{\text{HSC + Tgfb + DCN}}$ across the data points of biological replicates and reported the *p*-value in the figure.

## Benchmarking NiCo cell type niche interactions with tissue niche detection methods

To compare NiCo predictions for cell-cell interactions derived from logistic regression coefficients with topological tissue niche domains predicted by alternative methods, we define two scoring schemes:

**Niche composition Z-score.** For a given tissue domain $d$, we compute the frequency $f_{dj}$ of cell type $j$, and mean $\mu_j$ and standard deviation $\sigma_j$ of cell type $j$ across all domains. From these numbers, we compute a

z-score for the enrichment of cell type $j$ in domain $d$:

$$Z_{dj} = \frac{f_{dj} - \mu_j}{\sigma_j} \qquad (10)$$

**Cellcharter neighborhood enrichment score (CC enrichment).** As alternative scoring metric we compute the function cc.gr.enrichment from the CellCharter method. cc.gr.enrichment takes an anndata object along with 'group_key' and 'label_key' as input and computes the enrichment of label_key in group_key. The function returns the ratio of observed to expected values and indicates whether a cell type is over- or under-represented in a given tissue domain.

We compare the cell type composition between the ground truth (GT) domains and method-detected domains using the Z-score and the CC enrichment. In both scenarios, we compute the Pearson correlation of each of these quantities between a GT domain and all method-detected domains and consider the method-detected domains with the highest correlation as best match. We then compare the distribution of maximum correlations for all GT domains across methods.

**Niche-detection benchmarking on Allen Brain spatial MERFISH atlas data and STARmap mouse visual cortex data.** We used the default parameters for CellCharter, Banksy, SpaGCN, SpatialPCA, SeuratV5, and Stagate, except for the clustering resolution parameter or the required number of clusters. We adjusted this parameter to retrieve a number of predicted domains close to the number of GT domains (6). For the Allen Brain MERFISH data, CellCharter identified 6 clusters that maximized the stability parameter. Banksy, with the default Leiden clustering resolution of 0.9 and lambda of 0.8, identified 15 clusters, which exceeds the ground truth (GT) of 6 domains. We also tested another clustering resolution of 0.2 with a lambda of 0.8, which resulted in 9 clusters. SpaGCN, with the default Louvain parameter, identified 23 clusters, while resolutions of 0.2, 0.1, and 0.05 identified 13, 9, and 3 clusters, respectively. For SpatialPCA, we specified the clusternum parameter as 6 at the louvain_clustering step. Similarly, in SeuratV5, we set the niches.k parameter to 6 in the BuildNicheAssay step. Stagate, with the default resolution of 1, identified 22 clusters; resolution of 0.5, 0.2, and 0.1 identified 15, 9, and 4 clusters, respectively.

After adjustment of clustering resolution or cluster number, all methods identified 7 clusters for STARmap mouse visual cortex data, equal to the number of GT domains, except for CellCharter, which identified only five clusters. All methods were used with default parameters except for the clustering resolution parameter or the required number of clusters. We set the niches.k parameter to 7 for Seurat and clusternum parameter to 7 for SpatialPCA. We used Leiden clustering with a resolution of 0.5 in Stagate, a resolution of 0.9 in Banksy, and a resolution of 1 in SpaGCN. CellCharter identified 5 clusters from stability analysis that were used to define the tissue domains.

**Cell-cell interaction benchmarking on Intestinal data**
We benchmarked Niche-DE[56], COMMOT[21], cellNeighborEX[58], and stLearn[57] on the intestinal data to predict ligand-receptor pairs enriched in pairs of interacting cell types. Niche-DE[56] identified 6 genes (*Mptx2, Slc12a2, Maoa, Clca3b, Apob, Stmn1*) between Paneth niche cells and the stem/TA index cell type, and 4 genes (*Stmn1, H2-Eb1, Maoa, Slc12a2*) between stem/TA niche cells and the Paneth index cell. CellNeighborEX[58] identified *Mptx2, Lgr5, and Kit* as cell-contact dependent differentially expressed genes between stem/TA and Paneth cells.

COMMOT[21] using default parameters that filter LR pairs where both ligand and receptor are expressed in at least 5% of the spots, identified 0 pairs with the CellChat database and 5 pairs (*Apob-Sdc1, Cadm1-Cadm1, Cd14-Itgam, Vcan-Cd44, Vim-Cd44*) with the NiCoLRdb

database. stLearn[57] detected five pairs of LR interactions (*Vim-Cd44, Vcan-Cd44, Cadm1-Cadm1, Gzmb-Chrm3, Cd34-Sell*) using the connectomeDB2020 database.

## Reporting summary
Further information on research design is available in the Nature Portfolio Reporting Summary linked to this article.

## Data availability
All datasets used in this paper are publicly available. Detailed pre-processing steps and cell type annotations are described in the "Dataset Description" section of the Methods. The scRNA-seq tissue data are available from the following sources: the mouse organogenesis data are available at ArrayExpress under accession number E-MTAB-6967; the mouse intestinal data are available at GEO under accession number GSE190037; the mouse liver cell atlas data (StSt) are available at https://www.livercellatlas.org/download.php; the mouse primary motor cortex data are available at https://assets.nemoarchive.org/dat-ch1nqb7.

The spatial datasets are available from the following sources: the mouse organogenesis seqFISH data are available at https://content.cruk.cam.ac.uk/jmlab/SpatialMouseAtlas2020/; the mouse small intestine MERFISH data are available at https://datadryad.org/stash/dataset/doi:10.5061/dryad.jm63xsjb2; the mouse liver MERFISH data are available at the Vizgen MERSCOPE website https://info.vizgen.com/mouse-liver-access/; the mouse primary motor cortex (coronal slice 153, mouse1) MERFISH data aren available at https://download.brainimagelibrary.org/29/3c/293cc39ceea87f6d/processed_data/. The cerebellum Slide-seqV2 data "myRCTD_cer_reps.rds" file was obtained using the scripts at https://raw.githack.com/dmcable/spacexr/master/AnalysisCSIDE/Figures/figure3.html and converted into a scanpy object that is available under the NiCo tutorial link https://nico-sc-sp.readthedocs.io/en/latest/tutorial2.html. The visual cortex STARmap data are available at https://github.com/prabhakarlab/Banksy_py/tree/main/data/starmap. The MERFISH Brain Atlas data C57BL6J-638850-raw.h5ad were downloaded from https://alleninstitute.github.io/abc_atlas_access/descriptions/MERFISH-C57BL6J-638850.html. The intestinal stem cell lineage scRNA-seq data are available at GEO under accession number [GSE94092]. The Supplementary Fig. source data are available at https://github.com/ankitbioinfo/nico_tutorial/tree/main/Supplementary_data. The processed source data, simulation datasets, benchmark data, and experimental data presented in the main figures are available at https://zenodo.org/records/13895622.

## Code availability
The open-source code for NiCo is available at a PyPI repository (https://pypi.org/project/nico-sc-sp/) and on Zenodo (https://zenodo.org/records/13902981)[99]. We uploaded all codes and scripts used for this study's analyses and figure plotting functions to the NiCo package. The documentation, usage instructions and other relevant information related to NiCo can be accessed through the tutorial (https://github.com/ankitbioinfo/nico_tutorial) and documentation (https://nico-sc-sp.readthedocs.io/en/latest/).

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

## Acknowledgements

We thank the members of the Grün lab for their advice and encouragement throughout this project. We also thank Julio Saez-Rodriguez, Jovan Tanevski, Moritz Lampert and Ingo Scholtes for helpful discussions and suggestions. We thank Carlos Talavera-Lopéz for critical reading and feedback on the manuscript. This work was supported by the German Research Foundation (DFG) (SPP1937 GA 2129/2-2, SFB1425 Project #422681845 and INST 93/1072-1 Project #471222118, to D.G.), by the CZI Seed Networks for the Human Cell Atlas (to D.G.), by the ERC (818846 - ImmuNiche - ERC-2018-COG, to D.G.), and by the Bundesministerium für Bildung und Forschung (BMBF) (TissueNet – 031L0311A and CureFib – 01EJ2201C, to D.G.).

## Author contributions

D.G. conceived the project and proposed the algorithm. D.G. and A.A. designed the algorithm. A.A. implemented the algorithm, collected and analyzed all data. S.B. performed in vitro culture experiments. S.T. carried out smFISH experiments. D.G. and A.A. wrote the manuscript. D. G. supervised the project. All authors reviewed the manuscript.

## Funding

## Competing interests

D.G. serves on the scientific advisory board of Gordian Biotechnology. The remaining authors declare no competing interests.
