## [Transparent Peer Review file · Nature Communications]

NiCo Identifies Extrinsic Drivers of Cell State Modulation by Niche Covariation Analysis

Corresponding Author: Professor Dominic Gruen

This manuscript has been previously reviewed at another journal. This document only contains reviewer comments, rebuttal and decision letters for versions considered at Nature Communications.

Version 0:

Reviewer comments:

Reviewer #1

(Remarks to the Author)

The authors have addressed all of our requests, and I am happy to recommend the manuscript for publication.

Reviewer #3

(Remarks to the Author)

Agrawal and colleagues describe NiCo, a computational framework to identify correlated transcriptomic programmes in co-localized cell types. In this revised manuscript, the authors improved the clarity and correctness of the manuscript, reported additional comparisons to several tools for the intermediate steps of their method, and conducted an additional experiment to confirm parts of a mechanism suggested by applying NiCo to mouse live samples. Significant efforts have been made to address the reviewers concerns which results in a stronger manuscript. In particular, many practical questions that readers would have are now answered. The approach is sound and clearly defined. However, some limitations of the original work, discussed thereafter, still remain.

1) As clearly stated by the authors, this tool is designed for identifying correlations between transcriptomic signatures in adjacent cell types, and supports spatial transcriptomics technologies that are currently widely used. Here my personal view diverges from other reviews as I believe that, independently from the future adoption of single-cell resolution spatial technologies with a complete transcriptome readout, there is value in being able to analyse the data currently being generated and that will inevitably still be produced in years to come. It is however correct to emphasize that NiCo is designed for a specific generation of spatial technologies, which limits its potential use cases. Moreover, as NiCo only models a specific type of spatial dependencies it might be of use to a restricted audience. Nevertheless, the tool has its well-defined scope and could be relevant for some applications.

2) The observations primarily focus on known processes identified by NiCo, among many other unexplored relationships. Spelling out how to systematically interpret the results provided by NiCo and derive new insight would make a very compelling argument. While the example on stellate and Kupffer cells is interesting and partly alidated externally, it is currently presented in way that is hard generalize as it starts by selecting the stellate cell niche based on its unremarkable predictability. The inactivation of TGF beta by binding of DCN was previously studied (PMID: 29435195) and evidence on stellate cell activation was reported (PMID: 17067743). The involvement of Tgfb1-expressing Kupffer cell, supported by the smFISH experiment, is novel to the best of my knowledge, but the reduction of TGF beta activation by Decorin seems to be incremental evidence for a pre-existing hypothesis. Clarifying which aspects are suggested by NiCo and validated and which are rediscovering known phenomena is important to allow the readership to properly judge the value of the findings.

3) From my perspective, the benchmarking on annotation and niches provided is sufficient for justifying the intermediate of a novel approach. More extensive work on defining the correct metric to assess the quality of zonation provided by different spatial analysis frameworks seems more critical to a systematic review than to the present manuscript. These comparisons also help to identify what differs between the approach proposed and existing methods, although it would be valuable to comment on conceptual and applicability differences with these tools rather only stating differences in performance. The difference between actual ground truth in simulated data and annotations (for both cell types and niches) could be made clearer e.g. by referring to the latter as "gold standards", which would make it more obvious that they correspond to curated

prior knowledge that can be partially recapitulated, despite a perfect match not being expected from different unsupervised methods with different aims.

Some more minor observations:

- * Please clarify what you mean by "central cell type with the highest correlation" in EDF2-3 and Fig 2.
- * In the main text, it is now ambiguous that the downstream analyses on stellate cells are done on the complete sample and not the sub-region.
- * A few typos are still present, such as: Results: "across technical replicates", "We note, that", "between Tgfb1 in Dcn in colocalized pairs". EDF5: "in the a UMAP representation".
- * Thank you for providing the frequencies of colocalization between cell types in your response to my comments. I believe reporting these values (or their range across cell types) in the paper would provide readers with a concrete sense of how abundant a cell type interaction needed to be to selected in these examples.

Response to the reviewer comments

The reviewer comments are shown in black and our responses are highlighted in blue.

Reviewer #1:

Remarks to the Author:

The authors have addressed all of our requests, and I am happy to recommend the manuscript for publication.

Remarks on code availability:

The readme is clear and the tutorials are extensive, so documentation seems very good. I have not tried to run it myself.

We thank this reviewer for the constructive feedback on the manuscript, which helped us to improve our study.

Reviewer #3:

Remarks to the Author:

Agrawal and colleagues describe NiCo, a computational framework to identify correlated transcriptomic programmes in co-localized cell types. In this revised manuscript, the authors improved the clarity and correctness of the manuscript, reported additional comparisons to several tools for the intermediate steps of their method, and conducted an additional experiment to confirm parts of a mechanism suggested by applying NiCo to mouse live samples. Significant efforts have been made to address the reviewers concerns which results in a stronger manuscript. In particular, many practical questions that readers would have are now answered. The approach is sound and clearly defined. However, some limitations of the original work, discussed thereafter, still remain.

We thank the reviewer for acknowledging the improvement of our revised manuscript and for the additional constructive feedback.

1) As clearly stated by the authors, this tool is designed for identifying correlations between transcriptomic signatures in adjacent cell types, and supports spatial transcriptomics technologies that are currently widely used. Here my personal view diverges from other reviews as I believe that, independently from the future adoption of single-cell resolution spatial technologies with a complete transcriptome readout, there is value in being able to analyse the data currently being generated and that will inevitably still be produced in years to come. It is however correct to emphasize that NiCo is designed for a specific generation of spatial technologies, which limits its potential use cases. Moreover, as NiCo only models a specific type of spatial dependencies it might be of use to a restricted audience. Nevertheless, the tool has its well-defined scope and could be relevant for some applications.

We appreciate that the reviewer recognizes the value of NiCo for the purpose it was designed for, i.e., the inference of cell state covariation driven by local niche interactions. However, we would like to note that applicability of NiCo is not restricted to a specific generation of spatial technologies. As we demonstrated by running NiCo on Slide-seqV2 data, NiCo can detect spatial cell state covariations in data generated with genome-wide spatial technologies without requiring reference scRNA-seq data. We added a tutorial to the online repository to provide guidance for this application.

In the revised discussion, we make the point clearer:

“With technological advancements, sequencing-based methods may be able to achieve single-cell resolution in the future, and NiCo will be directly applicable to this data type. Similarly, NiCo can be applied to future generations of imaging-based spatial transcriptomics technologies without requiring reference scRNA-seq data.”

2) The observations primarily focus on known processes identified by NiCo, among many other unexplored relationships. Spelling out how to systematically interpret the results provided by NiCo and derive new insight would make a very compelling argument. While the example on stellate and Kupffer cells is interesting and partly validated externally, it is currently presented in way that is hard to generalize as it starts by selecting the stellate cell niche based on its unremarkable predictability. The inactivation of TGF beta by binding of DCN was previously studied (PMID: 29435195) and evidence on stellate cell activation was reported (PMID: 17067743). The involvement of Tgfb1-expressing Kupffer cell, supported by the smFISH experiment, is novel to the best of my knowledge, but the reduction of TGF beta activation by Decorin seems to be incremental evidence for a pre-existing hypothesis. Clarifying which aspects are suggested by NiCo and validated and which are rediscovering known phenomena is important to allow the readership to properly judge the value of the findings.

We acknowledge this important remark. The key novelty of NiCo lies in the inference of cell state dependencies across co-localized cell types and in detecting the underlying gene

expression programs. In the specific case of the stellate cell niche, we agree with the reviewer, that inactivation of Tgf- β by Dcn is a known phenomenon, and our in vitro experimental validation confirms this observation. The new aspect discovered by NiCo is the covariation of Dcn expression in stellate cells with the expression of Tgfb1 in Kupffer cells, which suggests a feedback mechanism to dampen stellate cell activation. We believe that the in situ validation of the covariation between Dcn in stellate cells and Tgfb1 in co-localized Kupffer cells by smFISH provides evidence for this feedback mechanism. A conclusive in vivo validation would require a targeted inactivation of Tgf- β in Kupffer cells which could lead to a higher baseline level of stellate cell activation in the normal liver. These experiments could be an important subject for future research to investigate the control of fibrosis in the homeostatic liver.

To clarify the novel aspect discovered with NiCo we added the following sentence to the results section:

“The key insight inferred by NiCo is the induction of Dcn expression in HSCs as a result of Tgf- β upregulation in co-localized Kupffer cells, supported by the validated covariation pattern (Fig. 6g).”

3) From my perspective, the benchmarking on annotation and niches provided is sufficient for justifying the intermediate of a novel approach. More extensive work on defining the correct metric to assess the quality of zonation provided by different spatial analysis frameworks seems more critical to a systematic review than to the present manuscript. These comparisons also help to identify what differs between the approach proposed and existing methods, although it would be valuable to comment on conceptual and applicability differences with these tools rather only stating differences in performance. The difference between actual ground truth in simulated data and annotations (for both cell types and niches) could be made clearer e.g. by referring to the latter as "gold standards", which would make it more obvious that they correspond to curated prior knowledge that can be partially recapitulated, despite a perfect match not being expected from different unsupervised methods with different aims.

We agree with the reviewer that it would be helpful to comment on conceptual and applicability differences between common spatial analysis methods used in the benchmarking and NiCo. To address this point, we added the following statement to the benchmarking results:

“We note that NiCo predicts global interaction coefficients as a basis for inferring cell state dependencies between interacting cell types, while domain detection methods predict local tissue regions with unique cell type compositions.”

We appreciate the reviewer’s remark on our use of the term “ground truths”. However, we clearly defined for each instance in the manuscript, what this term refers to, which makes it transparent for the reader. We agree that in some cases, such as cell type annotation, this may

not be the ground truth in absolute terms, but the definition makes it clear what it refers to. Introducing “gold standard” as an alternative term for reference annotation could be confusing, and therefore we would like to keep the term “ground truths” with proper definition in each case. We note that this terminology is very common in the literature. Although we would prefer to keep the term “ground truth”, we leave the final decision to the editor.

Some more minor observations:

* Please clarify what you mean by "central cell type with the highest correlation" in EDF2-3 and Fig 2.

We thank the reviewer for pointing out the lack of clarity, and rephrased the sentence:

“For NiCo, the cell type with the highest correlation of its regression coefficients to the Z-score/CC-score was selected.”

* In the main text, it is now ambiguous that the downstream analyses on stellate cells are done on the complete sample and not the sub-region.

We now specified that NiCo was applied to the full slice data for the reported covariation analysis.

* A few typos are still present, such as: Results: "across techinal replicates", "We note, that", "between Tgfb1 in Dcn in co-localized pairs". EDF5: "in the a UMAP representation".

Thanks for spotting. Corrected.

* Thank you for providing the frequencies of colocalization between cell types in your response to my comments. I believe reporting these values (or their range across cell types) in the paper would provide readers with a concrete sense of how abundant a cell type interaction needed to be to selected in these examples.

We added this information to the results.

Remarks on code availability:

Thank you again for openly providing the code to run NiCo together with example notebooks.

We thank the reviewer for testing our code.